# Learning Query-Aware Budget-Tier Routing for Runtime Agent Memory

**Haozhen Zhang** [* 1]   **Haodong Yue** [* 2]   **Tao Feng** [3]   **Quanyu Long** [1]   **Jianzhu Bao** [1]   **Bowen Jin** [3]   **Weizhi Zhang** [4]
**Xiao Li** [5]   **Jiaxuan You** [3]   **Chengwei Qin** [6]   **Wenya Wang** [1]

## Abstract

Memory is increasingly central to Large Language Model (LLM) agents operating beyond a single context window, yet most existing systems rely on offline, query-agnostic memory construction that can be inefficient and may discard query-critical information. Although runtime memory utilization is a natural alternative, prior work often incurs substantial overhead and offers limited explicit control over the performance-cost trade-off. In this work, we present **BudgetMem**, a runtime agent memory framework for explicit, query-aware performance–cost control. BudgetMem structures memory processing as a set of memory modules, each offered in three budget tiers (i.e., LOW/MID/HIGH). A lightweight router performs budget-tier routing across modules to balance task performance and memory construction cost, which is implemented as a compact neural policy trained with reinforcement learning. Using BudgetMem as a unified testbed, we study three complementary strategies for realizing budget tiers: implementation (method complexity), reasoning (inference behavior), and capacity (module model size). Across LoCoMo, LongMemEval, and HotpotQA, BudgetMem surpasses strong baselines when performance is prioritized (i.e., high-budget setting), and delivers better accuracy–cost frontiers under tighter budgets. Moreover, our analysis disentangles the strengths and weaknesses of different tiering strategies, clarifying when each axis delivers the most favorable trade-offs under varying budget regimes. Code is available at https://github.com/ViktorAxelsen/BudgetMem

---

[*]Equal contribution   [1]Nanyang Technological University [2]Tsinghua University [3]University of Illinois Urbana-Champaign [4]University of Illinois Chicago [5]Sun Yat-sen University [6]The Hong Kong University of Science and Technology (Guangzhou). Correspondence to: Chengwei Qin <chengweiqin@hkust-gz.edu.cn>.

*Proceedings of the 43rd International Conference on Machine Learning*, Seoul, South Korea. PMLR 306, 2026. Copyright 2026 by the author(s).

## 1. Introduction

Memory has become a core component of modern Large Language Model (LLM) agents, enabling them to retain and reuse information beyond a single context window for long-horizon interaction, personalization, and knowledge-intensive reasoning (Hu et al., 2025). Nevertheless, most prior work centers on offline, query-agnostic memory construction, where past context is preprocessed, compressed, or indexed in a fixed manner without conditioning on the downstream query (Kang et al., 2025; Fang et al., 2025; Zhong et al., 2024; Xu et al., 2025; Packer et al., 2023; Chhikara et al., 2025). This "*build once, use always*" paradigm can be wasteful and brittle: it spends computation regardless of what a particular query needs, while potentially omitting crucial information for specific queries.

Instead, an intuitive alternative is on-demand memory extraction, where computation is triggered at runtime based on the current query. This flexibility comes at a cost: it pushes memory processing to runtime, making cost and latency first-class concerns. In practice, industrial LLM systems increasingly provide explicit, often *tiered* compute controls (e.g., "thinking" modes (Singh et al., 2025), reasoning levels (OpenAI, 2025), or heavier-model options (Anthropic, 2025)), reflecting the need to balance quality against runtime cost. This motivates a key question for agent memory: *how can we enable explicit and controllable performance–cost trade-offs for runtime memory extraction?*

Unfortunately, enabling performance-cost trade-offs for runtime agent memory is fundamentally challenging. Most existing trade-off mechanisms operate offline, while on-demand memory pushes these decisions to runtime, where each query raises a quality-cost choice. This exposes two core questions. First, *where should budgets be applied?* Existing systems often treat memory as a monolithic pipeline with a fixed compute setting, making trade-offs coarse and difficult to control (Kang et al., 2025; Zhong et al., 2024). This motivates defining an appropriate budgeting unit for runtime settings, i.e., which modular part(s) of the memory extraction process should be assigned budgets, so that computation can be controlled in a targeted and effective way. Second, *how should budgets be realized?* Prior work provides little systematic guidance on trade-offs for run-

time memory, often resorting to ad hoc cost reduction or simply increasing compute (Yan et al., 2025a). As a result, even after a budgeting scheme is specified, it remains unclear how to operationalize budget control, which design axes best capture meaningful trade-offs, and how these choices behave across different budget regimes. Addressing these questions requires moving beyond one-off heuristics or compute-heavy escalation toward a more systematic view of performance–cost control for runtime agent memory.

To address the above challenges, we propose **BudgetMem**, a runtime agent memory framework that enables explicit, controllable performance–cost trade-offs for on-demand memory extraction. BudgetMem views runtime memory extraction as a multi-stage modular pipeline and makes computation controllable at the module level. Specifically, BudgetMem standardizes how each module is invoked by exposing a common budget-tier interface, so that a learned router can select among budget tiers within modules while keeping the overall extraction structure fixed.

Building on this modular backbone, BudgetMem provides three budget tiers (i.e., LOW/MID/HIGH) for each module, offering different quality-cost trade-offs. We instantiate budget tiers through three complementary tiering strategies: *implementation* tiering (varying the module implementation), *reasoning* tiering (varying inference behavior), and *capacity* tiering (varying the module's model capacity). To navigate these tiered choices, BudgetMem employs a shared lightweight router that performs budget-tier routing as the query is processed: at each module, it selects a tier based on the available context (i.e., the query and intermediate module states). We train the router with reinforcement learning under a cost-aware reward that trades off task performance against memory extraction cost, forming controllable performance–cost behavior. This design yields a practical and controllable runtime memory system, while also enabling a unified comparison of different budget realization strategies.

In experiments on LoCoMo, LongMemEval, and HotpotQA, BudgetMem delivers strong gains over competitive baselines in performance-first (high-budget) settings, and exhibits clear performance–cost trade-off curves as budgets tighten. Moreover, our analyses disentangle the relative strengths of different budget tiering strategies, offering insights into which mechanisms provide the best returns under different budget regimes.

Our contributions are summarized as follows:

- We introduce **BudgetMem**, a modular runtime agent memory framework that enables explicit performance–cost control for on-demand memory extraction via budget-tiered modules.

- We propose budget-tier routing, learning a shared lightweight router with reinforcement learning to se-

lect budget tiers during extraction and further instantiate three complementary budget realization strategies (i.e., *implementation*, *reasoning*, and *capacity* tiering) within a unified framework.

- Experiments on LoCoMo, LongMemEval, and HotpotQA demonstrate strong performance and clear performance–cost trade-offs, and our further analyses provide valuable insights into when different strategies deliver the best returns across budget regimes.

## 2. Related Work

### 2.1. Memory-Augmented LLM Agents

Memory-augmented LLM agents typically maintain an external memory store to overcome finite context windows and support long-horizon use. A large portion of prior work emphasizes offline or ahead-of-time memory construction, where past interactions are periodically summarized/compressed and indexed, and later accessed via retrieval at query time (Zhong et al., 2024; Packer et al., 2023; Lee et al., 2024; Kang et al., 2025; Fang et al., 2025). Representative designs organize memories chronologically and hierarchically (e.g., event summaries and persona profiles) with information retrieval and heuristic update rules such as recency-based decay (Zhong et al., 2024; Kang et al., 2025). More recent work enriches memory with agentic updates and structure, e.g., constructing metadata-rich notes and linking them into graphs for scalable retrieval and evolution (Xu et al., 2025), or using LLM-based memory managers to retrieve similar entries and apply discrete operations (i.e., add, update, delete, and no-op), with variants that build structured memories such as knowledge graphs (Chhikara et al., 2025). Several methods further introduce learning-based memory management (often via reinforcement learning) to optimize memory operations using downstream task signals (Yan et al., 2025b; Wang et al., 2025). On-demand utilization approaches also push beyond "retrieve-then-answer" by invoking deeper planning over memories (Yan et al., 2025a).

Despite these advances, runtime memory remains costly and is rarely framed as explicit performance–cost control: most work relies on fixed pipelines or studies efficiency mainly in offline construction (Fang et al., 2025). In contrast, we focus on runtime memory extraction and systematically compare controllable performance–cost trade-offs under budgets.

### 2.2. Inference-Time Performance-Cost Trade-offs in LLM Systems

A growing body of work studies how to trade LLM quality against runtime cost and latency by exposing controllable "compute knobs". Existing approaches broadly fall into three categories. (i) *Algorithmic and systems-level* optimizations reduce inference cost without changing the task, including

faster decoding (Fu et al., 2024; Cai et al., 2024; Li et al., 2024b), early-exit or adaptive-depth inference (Schuster et al., 2022), pruning/sparsity (Ma et al., 2023; Frantar & Alistarh, 2023; Sun et al., 2023), quantization (Xiao et al., 2023a; Liu et al., 2024), and serving-side optimizations such as caching/batching and KV-cache efficiency for long contexts (Zhang et al., 2023; Xiao et al., 2023b; Ge et al., 2023; Li et al., 2024a). (ii) *Reasoning-level* controls vary inference behavior under different budgets, e.g., direct generation vs. chain-of-thought (Wei et al., 2022), self-refinement and reflection loops (Yao et al., 2022; Shinn et al., 2023; Madaan et al., 2023), or bounded deliberation via limits on steps, samples, or search (Wang et al., 2022). (iii) *Capacity-level* controls vary effective model capacity, including mixture-of-experts activation (Shazeer et al., 2017; Fedus et al., 2022) and distillation-based deployments (Gu et al., 2023; Agarwal et al., 2024), closely related to LLM routing across backends under budget constraints (Chen et al., 2024; Feng et al., 2024; Zhang et al., 2025; Jin et al., 2025).

While these directions offer practical mechanisms for controllable trade-offs, they have been less systematically explored in runtime agent memory, where computation is spent on memory extraction rather than answer generation. In this work, we take a first step toward bringing explicit performance–cost control to runtime agent memory by introducing a budget-tiered framework and studying complementary budget realization strategies in a unified setting.

## 3. Problem Setup and Method Overview

In this work, we study *runtime* agent memory extraction, where the system selectively processes raw historical records *at query time* to construct a compact memory for answering the current user query. Unlike offline pipelines that pre-compress or pre-structure the entire history, we keep past records *intact* and defer memory computation until a query arrives, avoiding irreversible information loss introduced by query-agnostic preprocessing. Before detailing the methodology, we briefly summarize the task setup from the perspectives of inputs and outputs.

**Inputs.** Let $\mathcal{H}$ denote the LLM agent's history, e.g., prior conversations, logs, or documents. We first perform a lightweight, task-agnostic segmentation of $\mathcal{H}$ into a chunk store $\mathcal{C} = \{c_i\}_{i=1}^N$, where each $c_i$ is a short text chunk[1]. Given a user query $q$, a retriever $\mathcal{R}$ returns a subset of potentially relevant chunks:

$$\mathcal{C}_q = \mathcal{R}(q, \mathcal{C}), \qquad \mathcal{C}_q \subset \mathcal{C}. \tag{1}$$

where $(q, \mathcal{C}_q)$ serves as input to runtime memory pipeline.

---

[1]This step only segments raw text for indexing and does not perform any offline extraction, summarization, or rewriting.

**Outputs.** The system produces (i) an extracted memory $m$ that is compact and useful for the query, and (ii) a final answer $\hat{y}$ generated by an LLM conditioned on $(q, m)$:

$$m = f_{\text{mem}}(q, \mathcal{C}_q), \qquad \hat{y} = f_{\text{ans}}(q, m). \tag{2}$$

Here $f_{\text{mem}}$ is a learned, budget-controlled memory extraction procedure (our focus), while $f_{\text{ans}}$ is an answer generator (e.g., a fixed LLM).

**Budgeted runtime extraction.** A key challenge is that runtime extraction occurs online and can be expensive; thus we seek explicit and controllable performance–cost trade-offs. To this end, we design a modular memory pipeline in which each module exposes a small set of *budget tiers* (LOW/MID/HIGH). A shared lightweight router makes *budget-tier routing* decisions as the query is processed, selecting which tier to use for each module based on the available context (the query and intermediate signals). The extracted memory produced by the final module is then used to answer the query, and the end-to-end process yields a task-level performance signal and an extraction cost signal for optimizing the router in an end-to-end manner.

## 4. BudgetMem

In this section, we first introduce the BudgetMem modular pipeline backbone (Sec. 4.1), then formalize budget tiers and present three tiering strategies (Sec. 4.2), and finally detail how we learn budget-tier routing with a lightweight router optimized via reinforcement learning (Sec. 4.3).

### 4.1. Modular Runtime Memory Pipeline

As shown in Figure 1, BudgetMem implements $f_{\text{mem}}$ as a *multi-stage modular pipeline* that progressively refines retrieved raw chunks into a query-focused memory. We instantiate the pipeline with a simple, interpretable module composition (filtering → parallel extraction → summarization) for our dialogue-centric evaluation, serving as a concrete backbone for studying budget control. Note that while we use this instantiation in our experiments, the framework itself does not assume a specific module set, and can be adapted to other modular memory pipelines.

**Pipeline structure.** Given $(q, \mathcal{C}_q)$, BudgetMem executes a fixed modular pipeline:

$$\mathcal{M}_{\text{fil}} \rightarrow \{\mathcal{M}_{\text{ent}}, \mathcal{M}_{\text{tmp}}, \mathcal{M}_{\text{top}}\} \rightarrow \mathcal{M}_{\text{sum}}. \tag{3}$$

At a high level, $\mathcal{M}_{\text{fil}}$ (i.e., filter module) refines the retrieved context, the parallel modules $\{\mathcal{M}_{\text{ent}}, \mathcal{M}_{\text{tmp}}, \mathcal{M}_{\text{top}}\}$ (i.e., entity, temporal, and topic module) extract complementary intermediate contexts, and $\mathcal{M}_{\text{sum}}$ aggregates them into the final memory. The output of an earlier module is fed into

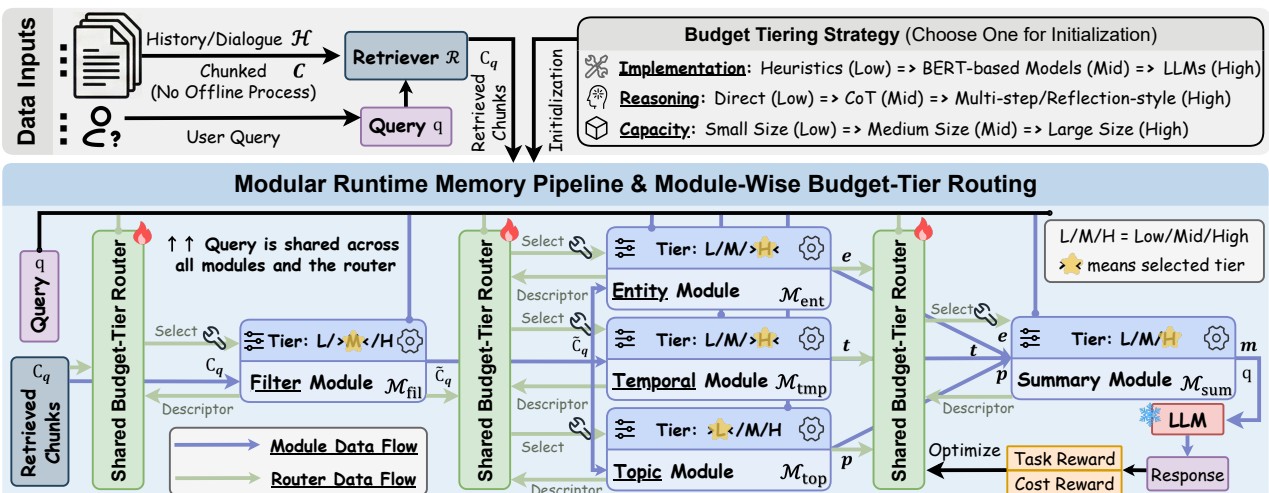

*Figure 1.* **BudgetMem overview.** Given a user query $q$, we retrieve raw chunks $\mathcal{C}_q$ from a chunked history (without offline memory preprocessing) and process them with a modular pipeline (filter → entity/temporal/topic → summary). Each module exposes LOW/MID/HIGH budget tiers instantiated by one of three strategies (**implementation**, **reasoning**, **capacity**). A shared lightweight router selects tiers module-wise based on the query and intermediate states, and is trained with reinforcement learning using task and cost rewards to yield controllable performance–cost trade-offs. ($\mathcal{C}_q$: retrieved raw chunks, $\tilde{\mathcal{C}}_q$: filtered chunks, $e, t, p$: extracted contexts, $m$: extracted memory)

later modules, forming intermediate states that the pipeline refines over stages. Since our focus is on budget-tier control and routing rather than the specific module implementations, we leave module details to Appendix A.5.

**Module interfaces and intermediate states.** We represent each module invocation by a context that includes the query and the currently available signals. The filtering module produces a refined chunk set

$$\tilde{\mathcal{C}}_q = \mathcal{M}_{\text{fil}}(q, \mathcal{C}_q). \tag{4}$$

where $\tilde{\mathcal{C}}_q$ can be viewed as a more focused subset or reweighted set of chunks. Next, three extraction modules operate in parallel on the filtered context:

$$e = \mathcal{M}_{\text{ent}}(q, \tilde{\mathcal{C}}_q), \quad t = \mathcal{M}_{\text{tmp}}(q, \tilde{\mathcal{C}}_q), \quad p = \mathcal{M}_{\text{top}}(q, \tilde{\mathcal{C}}_q). \tag{5}$$

where $e, t, p$ denote extracted contexts (e.g., entity-centric facts, temporal cues, topical summaries) that can be represented as text, structured fields, or compact natural-language notes. Finally, the summarization module aggregates these outputs into the extracted memory:

$$m = \mathcal{M}_{\text{sum}}(q, e, t, p), \tag{6}$$

which is then provided to the answer generator (i.e., $\hat{y} = f_{\text{ans}}(q, m)$).

**Budget-tier interface (conceptual).** Each module $\mathcal{M}$ in the pipeline is exposed through a common *budget-tier interface*, meaning that the module can be invoked under

different computation budgets (LOW/MID/HIGH) while preserving the same input–output contract. This standardization enables module-level budget control and lays the foundation for learning budget-tier routing (Sec. 4.3) and for comparing different tiering strategies (Sec. 4.2).

### 4.2. Budget Tiers and Tiering Strategies

BudgetMem equips each module $\mathcal{M}$ in the pipeline with a small set of *budget tiers* to expose explicit quality–cost options during runtime extraction. Specifically, every module provides three tiers, LOW/MID/HIGH, which correspond to increasing computational budget and (typically) improved extraction quality. Importantly, budget tiers are defined within each module via a common budget-tier interface (Sec. 4.1), enabling the router to make fine-grained, module-wise budget decisions without altering the pipeline structure.

**Tiering strategies.** A key question is how to realize these tiers in practice. Motivated by common performance–cost knobs used in modern LLM systems (Sec. 2.2), we study three complementary tiering strategies that capture different sources of cost variation: (i) **implementation tiering**, which varies the *module implementation* (from lightweight heuristics to learned task-specific models to LLM-based processing); (ii) **reasoning tiering**, which varies the *inference behavior* while keeping the underlying model backbone fixed (e.g., direct generation vs. more deliberative inference patterns); and (iii) **capacity tiering**, which varies the *model capacity* used inside a module (e.g., smaller vs. larger LMs). These strategies are orthogonal: they trade computation through different mechanisms (algorithmic procedure, rea-

soning patterns, and model size), enabling direct comparison of their trade-off characteristics in a unified framework.

**Realizing LOW/MID/HIGH.** For **implementation tiering**, LOW uses lightweight rule-based or pattern-based processing, MID uses a compact learned model specialized for the module function (typically BERT-based models (Devlin et al., 2019)), and HIGH upgrades the module to LLM-based processing for higher-quality extraction. For **reasoning tiering**, tiers are realized by progressively more compute-intensive inference behaviors (e.g., LOW: direct; MID: CoT-style (Wei et al., 2022); HIGH: multi-step/reflection-style (Shinn et al., 2023)). For **capacity tiering**, tiers correspond to increasing model sizes used to implement the same module function. Concrete instantiations for each module under each strategy are provided in Appendix A.5.

### 4.3. Learning Module-Wise Budget-Tier Routing

Given tiered modules, BudgetMem learns a shared lightweight router that performs *budget-tier routing* throughout the modular pipeline. As the query is processed stage-by-stage, the router selects a budget tier for each module invocation, yielding a sequence of tier decisions for a single query. Since the memory extraction process can involve non-differentiable components, we formulate routing as a sequential decision problem and optimize the router with reinforcement learning.

**Policy, state, and action.** Let $\pi_\theta$ denote the router policy with parameters $\theta$. At each module invocation step $k$, the router observes a state $s_k$ that summarizes the available context, and outputs an action $a_k \in \{\text{LOW}, \text{MID}, \text{HIGH}\}$ selecting the tier for the current module. Concretely, we construct $s_k$ from: (i) the query $q$, (ii) the current module input (i.e., the output from the previous module), and (iii) a module descriptor indicating which module is being routed[2]. The chosen action $a_k$ determines the budget tier at which the module is executed, producing the next intermediate signal(s) and thus the next state.

**Episode and objective.** A complete run of the modular pipeline for one query constitutes an episode. After executing all routed modules, we obtain an extracted memory $m$ and generate an answer $\hat{y} = f_{\text{ans}}(q, m)$, from which we compute a task performance reward $r_{\text{task}} \in [0, 1]$. In addition, the routed module executions incur an extraction cost, which we convert into a cost reward $r_{\text{cost}}$. We optimize the router under a cost-aware objective that trades off task performance and extraction cost:

$$r = r_{\text{task}} + \lambda \cdot \alpha \cdot r_{\text{cost}}, \quad (7)$$

where $\lambda$ controls the performance–cost trade-off preference, and $\alpha$ is a scaling factor used to align reward magnitudes (described below). We use a standard policy optimization algorithm (e.g., PPO (Schulman et al., 2017)) to optimize $\pi_\theta$; algorithmic details are provided in Appendix A.4.

**Cost modeling.** We define the raw extraction cost as the sum of per-module costs along the routed pipeline:

$$c_{\text{raw}} = \sum_k c(\mathcal{M}_k, a_k). \quad (8)$$

For LLM-based tiers, we measure $c(\cdot)$ by token usage multiplied by the corresponding input/output token prices; for non-LLM tiers (e.g., lightweight heuristics), the cost is treated as negligible in comparison. To make the cost term commensurate with $r_{\text{task}}$, we apply a sliding-window normalization to map costs to a bounded scale (Zhang et al., 2025). Specifically, we maintain a recent window of raw costs and normalize $\sqrt{c_{\text{raw}}}$ using robust quantiles:

$$\tilde{c} = \frac{\sqrt{c_{\text{raw}}} - Q_5}{Q_{95} - Q_5}, \qquad r_{\text{cost}} = 1 - \text{clip}(\tilde{c}, 0, 1), \quad (9)$$

where $Q_5$ and $Q_{95}$ denote the 5th and 95th percentiles computed over the sliding window.

**Reward-scale alignment.** In practice, we observe that $r_{\text{task}}$ and $r_{\text{cost}}$ can exhibit different variances during training, which may cause the higher-variance term to dominate policy updates. To mitigate this, we introduce a simple variance-based alignment factor:

$$\alpha = \frac{\text{std}(r_{\text{task}})}{\text{std}(r_{\text{cost}}) + \epsilon}, \quad (10)$$

where $\text{std}(\cdot)$ is computed over recent training rewards and $\epsilon$ is a small constant. The final reward in Eq. 7 then balances (i) the controllable trade-off via $\lambda$ and (ii) stabilized optimization via $\alpha$.

## 5. Experimental Setup

### 5.1. Datasets, Metrics, and Baselines

**Datasets.** We evaluate BudgetMem on three benchmarks that stress long-horizon memory and long-context question answering (QA): **LoCoMo** (Maharana et al., 2024) and **LongMemEval** (Wu et al., 2024), which are widely used for agent memory evaluation, and **HotpotQA** (Yang et al., 2018)[3] as a representative long-context QA task.

**Metrics.** We report both task performance and cost to characterize performance–cost trade-offs. For task performance, we use F1-score (F1) and LLM-as-a-judge (Judge)

---

[2]We implement the state as a compact embedding of these components; details are deferred to Appendix A.4.

[3]we follow the evaluation protocol adopted in prior work (Yu et al., 2025)

*Table 1.* **Experimental results on LoCoMo, LongMemEval, and HotpotQA datasets using F1-score (F1), LLM-as-a-Judge (Judge), and Cost under the *performance-first* setting.**

| Methods | LoCoMo | | | LongMemEval | | | HotpotQA | | | Avg. | | |
|---|---|---|---|---|---|---|---|---|---|---|---|---|
| | F1 | Judge | Cost$^\downarrow$ | F1 | Judge | Cost$^\downarrow$ | F1 | Judge | Cost$^\downarrow$ | F1 | Judge | Cost$^\downarrow$ |
| **LLaMA-3.3-70B-Instruct** | | | | | | | | | | | | |
| ReadAgent | 22.48 | 31.05 | 0.57 | 20.75 | 27.72 | 13.68 | 15.33 | 30.08 | 4.19 | 19.52 | 29.62 | 6.14 |
| MemoryBank | 22.27 | 28.98 | 0.73 | 26.74 | 32.67 | 3.94 | 22.25 | 23.75 | 7.75 | 23.75 | 28.47 | 4.14 |
| A-MEM | 26.43 | 32.96 | 2.88 | 21.74 | 33.17 | 80.02 | 43.25 | 54.69 | 26.74 | 30.47 | 40.27 | 32.07 |
| LangMem | 22.31 | 25.96 | **0.48** | 12.00 | 17.00 | 16.60 | 22.78 | 22.66 | 10.95 | 19.03 | 21.87 | 9.34 |
| Mem0 | 11.04 | 28.18 | 2.89 | 27.70 | 42.08 | 13.57 | 28.03 | 36.72 | 4.30 | 22.26 | 35.66 | 6.92 |
| MemoryOS | 30.62 | 34.55 | 1.97 | 12.97 | 33.50 | 38.83 | 34.50 | 43.36 | 13.32 | 26.03 | 37.14 | 18.04 |
| LightMem | 33.88 | 40.76 | 1.50 | 26.74 | 48.51 | 5.28 | 45.73 | 58.37 | 10.10 | 35.45 | 49.21 | 5.63 |
| **BudgetMem-IMP** | 38.75 | 50.32 | 1.80 | 37.47 | 56.00 | 0.71 | 49.31 | **65.77** | 1.35 | 41.84 | 57.36 | **1.29** |
| **BudgetMem-REA** | 40.92 | 52.23 | 2.90 | **40.53** | 58.00 | **0.67** | 51.12 | 61.93 | 0.99 | 44.19 | 57.39 | 1.52 |
| **BudgetMem-CAP** | **43.05** | **54.62** | 2.40 | 40.24 | **60.50** | 0.80 | **53.87** | 64.85 | **0.93** | **45.72** | **59.99** | 1.38 |
| **Qwen3-Next-80B-A3B-Instruct**[†] | | | | | | | | | | | | |
| ReadAgent | 22.45 | 31.37 | 0.24 | 27.72 | 20.75 | 4.97 | 18.05 | 25.78 | 1.75 | 22.74 | 25.97 | 2.32 |
| MemoryBank | 23.53 | 34.71 | 0.25 | 10.56 | 28.22 | 3.45 | 18.64 | 31.25 | 1.79 | 17.58 | 31.39 | 1.83 |
| A-MEM | 27.65 | 38.54 | 2.88 | 10.82 | 31.19 | 21.00 | 40.54 | 50.39 | 8.34 | 26.34 | 40.04 | 10.74 |
| LangMem | 20.89 | 23.40 | **0.14** | 11.01 | 14.00 | 3.99 | 20.77 | 21.09 | 17.56 | 17.56 | 19.50 | 2.82 |
| Mem0 | 10.77 | 25.32 | 1.15 | 25.71 | 36.14 | 4.96 | 24.72 | 37.89 | 2.02 | 20.40 | 33.12 | 2.71 |
| MemoryOS | 35.43 | 38.85 | 0.75 | 13.35 | 33.00 | 15.84 | 41.21 | 53.52 | 11.68 | 30.00 | 41.79 | 9.42 |
| LightMem | 32.85 | 42.83 | 0.70 | 27.70 | 47.52 | 3.39 | 41.29 | 55.42 | 8.56 | 33.95 | 48.59 | 4.21 |
| **BudgetMem-IMP** | 40.14 | **54.38** | 0.80 | 29.18 | 52.00 | 0.30 | 46.67 | 57.42 | 0.63 | 38.66 | 54.60 | 0.58 |
| **BudgetMem-REA** | 40.19 | 53.34 | 1.11 | **35.84** | **59.00** | 0.26 | 57.67 | 70.83 | **0.17** | **44.57** | **61.06** | 0.51 |
| **BudgetMem-CAP** | **41.22** | 53.18 | 0.61 | 31.01 | 56.00 | 0.17 | **58.70** | **72.08** | 0.22 | 43.64 | 60.42 | **0.33** |

**Bold** indicates the best score within each base model block. Avg. is averaged over datasets for each metric.
**Cost** is computed by summing all tokens across model calls and applying the model's respective input- and output-token prices
[†] indicates no training using this base model (transfer evaluation only).

to assess the correctness of a model prediction against the ground-truth answer. For cost, we measure memory extraction cost by aggregating API usage (input & output tokens) for all model calls made by BudgetMem during extraction, and converting token counts into monetary cost using the corresponding service pricing[4].

**Baselines.** We compare BudgetMem against a diverse set of strong memory-augmented baselines: (1) **ReadAgent** (Lee et al., 2024), (2) **MemoryBank** (Zhong et al., 2024), (3) **A-MEM** (Xu et al., 2025), (4) **LangMem** (LangChain, 2025), (5) **Mem0** (Chhikara et al., 2025), (6) **MemoryOS** (Kang et al., 2025), and (7) **LightMem** (Fang et al., 2025). These baselines cover a broad range of memory architectures, providing a comprehensive comparison against prior agent memory systems. More details are provided in Appendix A.

### 5.2. Implementation Details

We use LLaMA-3.3-70B-Instruct (Grattafiori et al., 2024) and Qwen3-Next-80B-A3B-Instruct (Yang et al., 2025) as the base LLMs, accessed via an API service. We train the

---

[4] https://www.together.ai/pricing

budget-tier router with LLaMA as the memory-extraction backbone, then directly test the same trained router with Qwen as the backbone without retraining. For the retrieval and chunking, we split each history or document into fixed-length text chunks (with a default chunk size of 256 for LoCoMo and LongMemEval; 1024 for HotpotQA) and use Contriever (Izacard et al., 2021) as the default retriever. For each query, we retrieve a candidate set of chunks and then apply the same top-$K$ budget ($K$=5) for downstream memory processing, so that BudgetMem and all baselines operate under a matched retrieval context size.

As for training, we follow a $6/2/2$ split for train/validation/test across datasets, and keep the same data splits and evaluation protocols for all methods to ensure fair comparison. We employ the Proximal Policy Optimization (PPO) (Schulman et al., 2017) as the default RL algorithm and use Adam optimizer with a batch size of 32 for up to 600 training steps. Appendix A and C provide additional hyperparameters, prompts, and implementation details.

## 6. Experimental Analysis

We organize experiments as follows. In Sec. 6.1, we report performance-first results ($\lambda = 0$) and compare

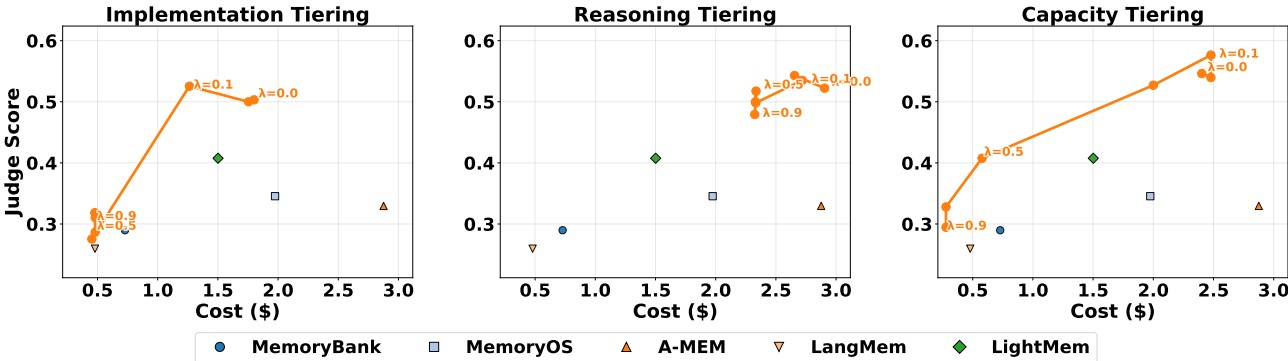

*Figure 2.* **Performance–cost trade-offs across tiering strategies on LoCoMo.** By varying the cost weight $\lambda$, BudgetMem traces smooth, controllable frontiers that shift toward higher performance as budget increases, and envelop baselines in both low- and high-cost regimes.

three variants: **BudgetMem-IMP**, **BudgetMem-REA**, and **BudgetMem-CAP**. Sec. 6.2 varies $\lambda$ to trace performance–cost curves, Sec. 6.3 ablates cost modeling and reward design, and Sec. 6.4 provides further analysis.

### 6.1. Main Results

As shown in Table 1, we evaluate BudgetMem under the performance-first setting on LoCoMo, LongMemEval, and HotpotQA, comparing against a diverse set of representative memory systems. The results support three key findings.

**Effectiveness.** BudgetMem delivers substantial improvements across all three datasets, consistently outperforming prior methods in both F1 and LLM-Judge. For example, on LongMemEval with the LLaMA-3.3-70B backbone, BudgetMem-CAP achieves a Judge score of 60.50, surpassing the strongest baseline LightMem (48.51) by a large margin, demonstrating stronger long-context evidence utilization and higher answer quality.

**Strong performance under controlled cost.** Even in the performance-first regime, BudgetMem remains cost-efficient, achieving higher quality without incurring excessive overhead. Notably, on HotpotQA with Qwen3-Next-80B-A3B, BudgetMem-CAP achieves the best Judge score (72.08) at a cost of 0.22, while BudgetMem-REA reaches a comparable Judge score (70.83) at an even lower cost (0.17). This efficiency stems from BudgetMem's runtime, on-demand design: rather than processing the entire history offline, it retrieves query-relevant raw chunks and spends extraction compute only when needed. We note an exception on LoCoMo, where histories are shorter and offline pipelines incur less overhead, so cost differences across methods become less pronounced.

**Strongest overall performance on aggregate.** When aggregating results across datasets, BudgetMem variants consistently rank at the top within each backbone block, indicating strong overall effectiveness rather than gains limited to a single benchmark. This trend holds for both LLaMA and Qwen, showing that BudgetMem delivers broadly strong performance across diverse evaluation settings under the performance-first regime.

### 6.2. Exploring Trade-off Across Tiering Strategies

As shown in Figure 2, we systematically compare the performance–cost distributions on LoCoMo across the three tiering axes. As the budget is relaxed (smaller $\lambda$), BudgetMem exhibits a smooth upward-leftward trend in the cost–performance, yielding a continuous and controllable trade-off frontier that envelopes prior baselines in both low- and high-cost regimes, achieving higher Judge at comparable cost or lower cost at comparable performance. Notably, *implementation* and *capacity* tiering span a broader cost range: implementation tiering delivers rapid Judge gains under moderate budgets, while capacity tiering continues to push the frontier outward as the budget increases, attaining the best quality in the high-budget regime. In contrast, *reasoning* tiering exhibits pronounced cost concentration: because it primarily adjusts inference behaviors (e.g., direct/reflection) while holding the underlying model capacity fixed, the additional cost is dominated by a relatively stable token overhead, leading to a narrower spread in cost. This pattern suggests that the reasoning axis acts more like a fine-grained quality knob within a limited cost bandwidth, providing meaningful improvements at similar cost, but offering less room for exploring extremely low-budget settings or extrapolating to very high-budget performance than implementation/capacity tiering. Overall, BudgetMem consistently advances the Pareto frontier from budget-constrained to performance-first regimes, and the differences among tiering axes further indicate that implementation/capacity tiering is better suited for widening budget coverage and extrapolating the boundary, whereas reasoning tiering is most effective for refining quality within a relatively concentrated cost region.

## 6.3. Ablating Reward-scale Alignment

As shown in Figure 3, we ablate the reward-scale alignment (Sec. 4.3) under the capacity tiering strategy on LoCoMo. Because the task reward and cost reward can differ markedly in scale and variability, removing alignment makes optimization unstable and can bias learning toward the cost term, yielding a degenerate low-cost policy. Empirically, without reward-scale alignment (and with a cost weight $\lambda=0.3$), the router collapses to selecting the Low tier for most modules, substantially reducing answer quality and driving the Judge score to the lowest level. In contrast, with reward-scale alignment enabled, the router learns a more graded use of tiers, producing a much smoother and better-behaved performance–cost frontier. This ablation highlights that reward-scale alignment is important for balancing learning signals, enabling meaningful cost–performance control rather than trivial cost minimization.

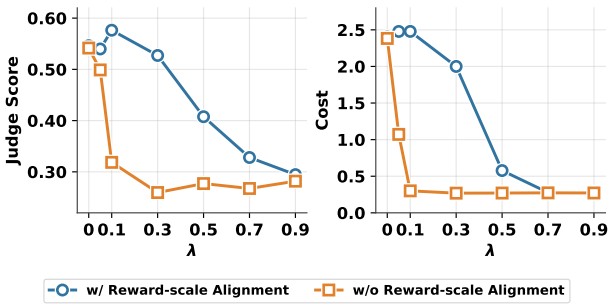

*Figure 3.* **Ablation of reward-scale alignment** under capacity tiering strategy on LoCoMo.

## 6.4. Discussion

**Budget tier selection ratio.** Figure 4 analyzes module-level routing behavior on LongMemEval using the capacity tiering strategy by reporting the selection ratios of LOW/MID/HIGH tiers under different cost weights $\lambda$. Overall, the router exhibits a clear and interpretable budget response: as cost pressure increases, it systematically shifts probability mass from higher-cost tiers to cheaper ones, providing module-level evidence that BudgetMem allocates computation in a cost-aware manner.

More concretely, with a small $\lambda$ (e.g., 0.1), the router assigns most modules to MID, prioritizing quality. At $\lambda = 0.3$, it increases the share of LOW while maintaining a substantial MID fraction, reflecting a controlled reduction in compute. Under larger $\lambda$, the policy further concentrates on LOW across modules to meet stricter cost preferences. Overall, these trends confirm that BudgetMem modulates tier usage in a predictable way as the budget preference changes.

**Retrieval-size sensitivity analysis.** Figure 5 shows cost and Judge score as we vary the number of retrieved raw

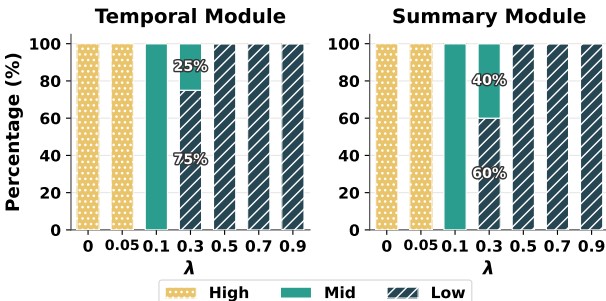

*Figure 4.* **Budget-tier selection ratios.** Module-wise LOW/MID/HIGH routing ratios on LongMemEval under varying cost weights $\lambda$ using the capacity tiering strategy.

chunks on LoCoMo, evaluated under all three tiering strategies. Increasing the retrieval size predictably raises cost due to longer inputs and additional processing, and it often improves Judge score by providing more supporting evidence, reflecting the standard trade-off between evidence coverage and computational overhead.

However, the benefit is not monotonic. In our setting, retrieving 5 chunks provides the best balance between cost and quality. Retrieving too many chunks can degrade performance: additional chunks introduce more redundant or weakly relevant content, increasing noise and potentially distracting the LLM, which lowers Judge score. Conversely, retrieving too few chunks provides insufficient evidence and limits downstream gains. Overall, these results highlight retrieval size as an important practical knob for balancing evidence sufficiency against context noise and cost.

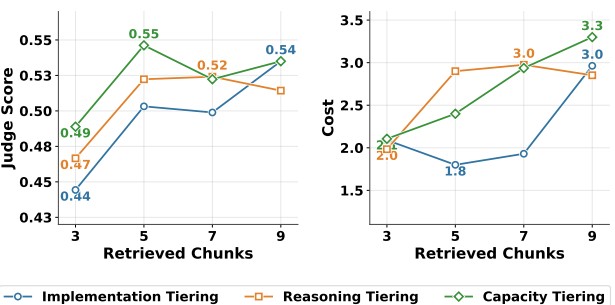

*Figure 5.* **Retrieval-size sensitivity on LoCoMo.** Cost and Judge versus the number of retrieved raw chunks, evaluated under all three tiering strategies.

**Latency analysis.** Although our main cost metric focuses on monetary cost, latency is also important for practical deployment. We therefore additionally measure latency under a controlled local deployment, using Qwen as the shared backbone and the same hardware and inference stack for all methods. We avoid API latency in this comparison because it can be strongly affected by network conditions and provider-side scheduling, making it less suitable for fair

relative comparison. Tables 2 and 3 report absolute latency in this setting, while we mainly use them to compare relative pipeline overhead across methods.

As shown in Table 2, BudgetMem exhibits the expected latency trade-off under different budget settings. For implementation tiering, increasing the cost weight reduces total inference latency from 3881 ms at $\lambda = 0$ to 1167 ms at $\lambda = 0.9$, while the retrieval/filter latency drops from 1176 ms to 46 ms. Reasoning and capacity tiering also reduce latency, although the reduction is milder because their tiers mainly change inference behavior or model capacity rather than replacing the filtering implementation. In addition, the measured GPU time remains small, around 44–52 ms per query, indicating that the lightweight router itself introduces limited overhead. As shown in Table 3, all methods benefit from batching, and the relative overhead of BudgetMem remains broadly consistent across serving settings. Overall, these results indicate that the retrieval/filter overhead is practically controllable, while system-level acceleration of the full pipeline is orthogonal to our goal and left for future work.

*Table 2.* **Detailed latency on LoCoMo under controlled local deployment.** Batch size is 1. All values are in milliseconds. **Off.** denotes average offline memory construction time, **Total** denotes average total inference latency, **Filt.** denotes average retrieval/filter module latency, and **GPU** denotes average GPU time. N/A indicates not applicable or not measured.

| Method | Off. | Total | Filt. | GPU |
|---|---|---|---|---|
| MemoryOS | 40657 | 2842 | 1488 | N/A |
| A-MEM | 26842 | 1449 | 49 | N/A |
| LightMem | 9740 | 1662 | 62 | N/A |
| Ours-IMP ($\lambda = 0$) | N/A | 3881 | 1176 | 46 |
| Ours-IMP ($\lambda = 0.3$) | N/A | 2432 | 556 | 46 |
| Ours-IMP ($\lambda = 0.9$) | N/A | 1167 | 46 | 46 |
| Ours-REA ($\lambda = 0$) | N/A | 3678 | 1274 | 47 |
| Ours-REA ($\lambda = 0.3$) | N/A | 3429 | 1216 | 45 |
| Ours-REA ($\lambda = 0.9$) | N/A | 3318 | 1209 | 48 |
| Ours-CAP ($\lambda = 0$) | N/A | 3461 | 1228 | 44 |
| Ours-CAP ($\lambda = 0.3$) | N/A | 3058 | 1079 | 46 |
| Ours-CAP ($\lambda = 0.9$) | N/A | 2853 | 976 | 52 |

*Table 3.* **Batch-size scaling on LoCoMo.** We report average total inference latency under different batch sizes. All values are in milliseconds. **BS** denotes batch size. For BudgetMem, we report Ours-REA with $\lambda = 0$.

| BS | MemoryOS | A-MEM | LightMem | Ours-REA |
|---|---|---|---|---|
| 1 | 2842 | 1449 | 1662 | 3678 |
| 2 | 1633 | 845 | 1059 | 2295 |
| 4 | 974 | 518 | 775 | 1327 |
| 8 | 604 | 296 | 438 | 858 |
| 16 | 387 | 154 | 269 | 547 |
| 32 | 203 | 96 | 163 | 326 |

## 7. Conclusion

We present **BudgetMem**, a runtime agent memory framework for explicit performance–cost control in on-demand memory extraction. BudgetMem equips each module in a modular memory pipeline with LOW/MID/HIGH budget tiers and learns a lightweight router to select tiers under a cost-aware objective. Using BudgetMem as a unified testbed, we compare three complementary tiering strategies—*implementation*, *reasoning*, and *capacity*—and characterize their trade-off behaviors across budgets. Experiments on LoCoMo, LongMemEval, and HotpotQA show strong performance in performance-first settings and improved performance–cost frontiers under tighter budgets, together with insights on when each strategy provides the best returns.

## Acknowledgements

This research/project is supported by the NTU Start-Up Grant (#023284-00001), Singapore, the MOE AcRF Tier 1 Seed Grant (RS37/24, #025041-00001), Singapore, and the Youth S&T Talent Support Programme of GDSTA (SKXRC2025462).

## Impact Statement

BudgetMem targets practical deployment challenges for memory-augmented agents by making memory computation explicitly controllable under budget constraints. This can benefit applications that require predictable latency or cost, and can lower barriers to using agent memory in resource-limited settings.

BudgetMem is an architectural and training framework and does not introduce new data sources beyond what memory-augmented agents already use. Potential risks are therefore similar to existing memory systems, such as occasional retrieval of irrelevant context or exposure of benign but unintended historical details. In practice, standard safeguards—e.g., sensible retention policies, basic access control, and routine evaluation of failure cases—can mitigate these issues. Overall, we expect BudgetMem's primary impact to be improving the usability and controllability of agent memory under real-world budget constraints.

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

# A. More Implementation Details

## A.1. Dataset Statistics

Table 4 summarizes the dataset statistics used in our experiments, including the train/validation/test split sizes (the number of training queries) and the average context length (in tokens) for LoCoMo (Maharana et al., 2024), LongMemEval (Wu et al., 2024), and HotpotQA (Yang et al., 2018).

Note that for HotpotQA, we follow the evaluation protocol and dataset construction in prior work (Yu et al., 2025): we concatenate evidence and distractor documents to form a long context, and use the same preprocessed test queries released in that setup.

*Table 4.* Dataset statistics.

| Dataset | #Train/Val/Test | Avg. context len. |
|---|---|---|
| LoCoMo (Maharana et al., 2024) | 1,236/446/304 | 18.0K tokens |
| LongMemEval (Wu et al., 2024) | 282/94/94 | 122.1K tokens |
| HotpotQA (Yang et al., 2018; Yu et al., 2025) | 5,250/1,750/128 | 26.0K tokens |

## A.2. Evaluation Details

Tables 5 and 6 report the evaluation configuration. Table 5 lists hyperparameters shared across all datasets. Table 6 further specifies dataset-dependent evaluation settings.

Unless otherwise specified, we repeat each BudgetMem experiment with three different random seeds and report the averaged results. This reduces the effect of random initialization and training stochasticity, especially for the reinforcement-learning-based router. We use the same data splits, retrieval settings, and evaluation protocol across runs to ensure a fair and reproducible comparison.

*Table 5.* Hyperparameter Settings (shared across all datasets)

| Hyperparameter | Value | Hyperparameter | Value |
|---|---|---|---|
| LLM Decoding Temperature | 0.0 | Batch Size | 32 |
| Embedding Dim (query/memory/desc) | 768 | Retrieval Top-$K$ | 5 |
| LLM-Judge Model | `openai/gpt-oss-120b` | | |
| Capacity Tiering Module Tier (LOW/MID/HIGH for Llama) | `L: meta/llama-3.2-3b-instruct;` | | |
| | `M: meta/llama-3.1-8b-instruct;` | | |
| | `H: meta/llama-3.3-70b-instruct` | | |
| Capacity Tiering Module Tier (LOW/MID/HIGH for Qwen) | `L: qwen/qwen2.5-7b-instruct;` | | |
| | `M: qwen/qwq-32b;` | | |
| | `H: qwen/qwen3-next-80b-a3b-instruct` | | |

*Table 6.* Evaluation hyperparameters (dataset-specific).

| Hyperparameter | LoCoMo | LongMemEval | HotpotQA |
|---|---|---|---|
| Chunking / chunk size | 256 | 256 | 1024 |
| Max Tokens for LLM Model | 32 | 512 | 512 |

## A.3. Router State Representation

We provide additional details on the router state representation used in Sec. 4.3 to improve reproducibility. For each module invocation, the router state is constructed from three textual components: the user query, the current module input, and a module descriptor indicating which module is being routed. We encode these three components separately using `all-mpnet-base-v2`. This produces one 768-dimensional embedding for the query, one 768-dimensional embedding for the module input, and one 768-dimensional embedding for the module descriptor.

We then concatenate the query embedding and the module-input embedding, and apply a linear projection to obtain a 256-dimensional context representation. The module descriptor embedding is projected separately into a 256-dimensional module representation. Finally, we concatenate the 256-dimensional context representation and the 256-dimensional module representation to form the final 512-dimensional router state vector. This state vector is fed into the shared budget-tier router to predict the tier selection distribution over LOW, MID, and HIGH for the current module. The embedding model is kept frozen during training, while the projection layers and the router policy network are optimized with reinforcement learning.

### A.4. Details on PPO Training Objectives

We train the shared budget-tier router policy $\pi_\theta$ with Proximal Policy Optimization (PPO) to make query-aware, module-wise tier selections along the fixed modular runtime memory pipeline: $M_{\text{fil}} \rightarrow \{M_{\text{ent}}, M_{\text{tmp}}, M_{\text{top}}\} \rightarrow M_{\text{sum}}$.

**Episode, actions, and states.** A single query constitutes one PPO episode: given $(q, C_q)$, the router sequentially selects a budget tier for each module invocation, executes the routed pipeline to obtain extracted memory $m$, and then produces the final answer $\hat{y} = f_{\text{ans}}(q, m)$. At each routing step $k$, the action $a_k$ chooses the module tier. We use a 3-way action space per module $\{\text{LOW}, \text{MID}, \text{HIGH}\}$. The state $s_k$ is a compact embedding of (i) the query $q$, (ii) the current module input (i.e., the intermediate output from the previous stage), and (iii) a *module descriptor* that indicates which module is being routed. We use a single unified actor–critic network shared across modules; the module identity is injected via the module-descriptor embedding, enabling parameter sharing while retaining module-specific routing behavior.

**Reward: performance–cost trade-off with normalization and alignment.** After finishing all routed module executions for a query, we compute a task reward $r_{\text{task}} \in [0, 1]$ from answer quality (F1 or LLM-judge), and a cost reward $r_{\text{cost}}$ from the extraction cost incurred by routed module calls. We optimize the cost-aware scalar reward as follows:

$$r \;=\; r_{\text{task}} + \lambda \cdot \alpha \cdot r_{\text{cost}}. \tag{11}$$

**Where.** $r_{\text{task}} \in [0, 1]$ is the task reward computed from answer quality (F1 or LLM-as-a-judge); $r_{\text{cost}}$ is the (normalized) cost reward derived from the extraction cost incurred by routed module calls; $\lambda$ controls the performance–cost trade-off; and $\alpha$ is the reward-scale alignment factor.

We compute $r_{\text{cost}}$ by applying sliding-window normalization to raw costs and mapping them to a bounded reward (Eq. 9), and compute $\alpha$ via variance-based reward-scale alignment (Eq. 10).

**PPO objective with joint (multi-module) likelihood ratio.** Let $\log \pi_\theta(a_k \mid s_k)$ be the per-module log-probability under the current policy. Because each query episode yields a sequence of module actions, we use the *joint* action probability over the routed trajectory:

$$\log \pi_\theta(\mathbf{a} \mid \mathbf{s}) \;=\; \sum_k \log \pi_\theta(a_k \mid s_k), \qquad \rho \;=\; \exp\Big( \log \pi_\theta^{\text{new}} - \log \pi_\theta^{\text{old}} \Big). \tag{12}$$

**Where.** At routing step $k$, $s_k$ is the router state embedding and $a_k \in \{\text{LOW}, \text{MID}, \text{HIGH}\}$ is the chosen budget-tier action for the current module; $\pi_\theta(\cdot)$ is the router policy parameterized by $\theta$ and $\log \pi_\theta(a_k \mid s_k)$ is the corresponding log-probability; $\mathbf{s} = \{s_k\}_k$ and $\mathbf{a} = \{a_k\}_k$ denote the full state–action sequence over a query episode, with joint log-likelihood $\log \pi_\theta(\mathbf{a} \mid \mathbf{s})$; and $\rho$ is the PPO likelihood ratio between the updated policy (superscript new) and the behavior policy used to collect rollouts.

We then apply the standard PPO clipped surrogate objective with ratio $\rho$, together with a value-function loss and an entropy bonus:

$$L \;=\; L_{\text{policy}} + c_v \, L_{\text{value}} - c_e \, H. \tag{13}$$

**Where.** $L_{\text{policy}}$ is the PPO clipped surrogate policy loss; $L_{\text{value}}$ is the value-function regression loss; $H$ is the policy entropy; and $c_v$ and $c_e$ are the coefficients for the value loss and entropy bonus, respectively.

We estimate advantages with a one-step Monte-Carlo return (the episode terminates after the routed pipeline completes), and use the critic as a baseline. For stability, we average entropy across module decisions (rather than summing), preventing the entropy term from scaling with the number of routed modules.

**Optimization and hyperparameters.** We use PPO as the default RL optimizer for router training. Unless otherwise specified, we use Adam with learning rate $3 \times 10^{-4}$, clip $\epsilon = 0.2$, $value$-loss coefficient $c_v = 0.5$, entropy coefficient $c_e = 0.01$, 4 PPO epochs per update, and gradient-norm clipping at $0.5$. We train with batch size 32 and up to 600 training steps. The router operates on 768-d embeddings (query/memory/descriptor), consistent with our shared hyperparameter settings.

### A.5. Module Implementation Details

**Unified budget-tier interface.** We follow the unified budget-tier interface: each module exposes three tiers (LOW/MID/HIGH). Budget tiers only alter the module's internal compute and inferential strength, enabling module-wise routing without changing the overall pipeline structure. To control for prompt effects across tiers, we standardize prompting by using each module's HIGH-tier prompt template consistently for LOW/MID/HIGH.

#### A.5.1. TIERING STRATEGY: IMPLEMENTATION TIERING

**Budgeted tiers with a fixed contract.** Implementation tiering realizes LOW/MID/HIGH by increasing the internal compute budget of a module while keeping its input–output contract unchanged. textscLow uses *cheap symbolic pipelines* (rules, regex, lightweight NLP such as spaCy) for fast and controllable processing; MID upgrades to a *compact learned specialist* (typically BERT-style encoders (Devlin et al., 2019)) that improves semantic modeling at moderate cost; HIGH further upgrades to *LLM-based processing* for the strongest open-ended reasoning and cross-context integration, at the highest cost.

**Filter Module.** The filter module realizes budget tiering through a monotonic upgrade of its internal relevance estimator, moving from inexpensive symbolic signals to semantic representations and finally to LLM-based evaluation. At LOW, it relies on sparse lexical/pattern matching with lightweight heuristic scoring, which is fast and deterministic and thus suitable for aggressive cost control. At MID, it introduces representation-based relevance (e.g., embedding- and BERT-style similarity), so that semantic matches beyond exact overlap can be recovered under a moderate compute budget. At HIGH, it further upgrades to LLM-based scoring over candidates, thereby enabling context-sensitive discrimination when relevance depends on implicit references, paraphrases, or multi-hop semantics; importantly, across tiers the module keeps the same output contract (a ranked top-$k$ candidate set), so improvements can be attributed to increased internal compute rather than interface changes.

**Entity Module.** The entity relation module follows the same budget ladder, but instantiates it for structured relational cue extraction. At LOW, it uses lightweight rules and shallow NLP cues (e.g., regex patterns and spaCy NER/dependency signals) to extract explicit relational traces with high controllability, although coverage is limited when relations are implicit or span multiple chunks. At MID, it replaces heuristics with a compact task-specific extractor—typically BERT-style relation extraction (e.g., REBEL-like triplet generation)—which improves normalization and coverage of subject–relation–object tuples under a moderate budget. At HIGH, it employs an LLM to perform open-domain relation completion and abstraction, so that missing links can be recovered and paraphrased relations can be consolidated into consistent tuples across chunks; nevertheless, the module continues to emit a standardized relation list, ensuring that downstream fusion remains unchanged.

**Temporal Module.** The temporal relation module is tiered as explicit cue detection → learned temporal extraction → generative temporal inference, aligning budget increases with progressively stronger temporal modeling. At LOW, it identifies explicit time expressions and constructs event–time anchors using lightweight NLP, which is reliable when temporal cues are directly stated. At MID, it adopts a compact learned extractor (e.g., BERT-style) to select temporally relevant relations and normalize them into a consistent form. At HIGH, it upgrades to LLM-based temporal reasoning, enabling implicit ordering, cross-chunk alignment, and global consistency checking when the timeline must be inferred rather than read off; across all tiers, the output contract is preserved as a standardized set of temporal facts/anchors.

**Topic Module.** The topic module implements budget tiering by progressively strengthening discourse-level abstraction from cheap lexical cues to learned representations and finally LLM-based topic reasoning. At LOW, it extracts controllable topical signals from inexpensive lexical/statistical cues (e.g., keywords, TF-IDF-like patterns, and simple transition markers), providing a low-cost but coarse description of topical content and shifts. At MID, it incorporates compact learned representations (e.g., BERT-style encoders and lightweight segment/topic modeling optionally assisted by NLP structure), which yields more stable topical units and more reliable transition detection under moderate compute. At HIGH, it uses an LLM to produce higher-level topic relations and evolution narratives, improving abstraction and interpretability for long or heterogeneous contexts; critically, each tier still outputs the same standardized topic-cue list so that downstream summarization can consume it uniformly.

**Summary Module.** The summary module realizes budget tiering by varying the depth of evidence fusion while keeping the final memory format unchanged. At LOW, it produces a deterministic evidence digest (e.g., concise aggregation/concatenation of retrieved memories and cues), minimizing compute and distortion. At MID, it outputs a structured synthesis that organizes key evidence, entities, temporal anchors, and topic cues into a parseable template, thereby improving usability and reducing brittleness without invoking expensive generative reasoning. At HIGH, it upgrades to LLM-based integrative summarization, where the model fuses multi-source cues, resolves conflicts, and generates an explanatory final memory representation that better supports complex queries; across tiers, the summary remains contract-compatible, so tier effects reflect the allocated compute budget rather than changes in downstream expectations.

### A.5.2. TIERING STRATEGY: INFERENCE BEHAVIOUR

**Budgeted inference via behaviourally tiered reasoning.** We realize *reasoning tiering* by allocating different compute budgets to a module through progressively stronger *inference behaviours*, while keeping the module interface fixed. Specifically, LOW executes *direct* inference; MID executes *CoT-style* inference; and HIGH executes *multi-step/reflection-style* inference that emphasizes iterative refinement and global consistency. Across tiers, each module consumes the same conditional inputs (the query and candidate memories or intermediate cues) and produces the same standardized artifact (e.g., a ranked subset or a cue list); therefore, tier differences reflect changes in internal reasoning depth and token footprint rather than changes in pipeline structure.

**Filter Module.** Across tiers, the filter module ranks retrieved chunks by relevance and returns the top-$k$. LOW performs direct scoring to maximize efficiency and determinism; MID adopts CoT-style scoring to improve semantic calibration beyond surface overlap; and HIGH adopts multi-step/reflection-style scoring to better resolve cases where relevance depends on implicit constraints or cross-memory interactions.

**Entity Module.** The entity module outputs the same relation list format across tiers; tiers only affect relation extraction strength. LOW extracts relations via direct inference and thus prioritizes precision on explicitly stated facts; MID uses CoT-style inference to improve coverage and normalization of relation tuples; and HIGH uses multi-step/reflection-style inference to consolidate paraphrases, reconcile competing candidates, and enforce relation consistency across memories.

**Temporal Module.** The temporal module always outputs the same set of temporal facts/anchors; tiers only change how well it infers and normalizes them. LOW identifies explicit anchors via direct inference; MID uses CoT-style inference to select and normalize query-relevant temporal constraints; and HIGH uses multi-step/reflection-style inference to improve cross-chunk alignment, infer implicit ordering, and promote global timeline coherence.

**Topic Module.** Across tiers, the topic module preserves the topic-cue list format, while producing progressively higher-level topic abstractions.LOW produces coarse topical cues via direct inference; MID uses CoT-style inference to yield more stable topic relations and transitions; and HIGH uses multi-step/reflection-style inference to improve abstraction quality and maintain global coherence over long and heterogeneous contexts.

**Summary Module.** Across tiers, the summary module preserves the final memory interface, while improving evidence aggregation and fusion. LOW performs direct synthesis with minimal inference; MID performs CoT-style synthesis that better organizes entity/temporal/topic cues into a coherent structure; and HIGH performs multi-step/reflection-style synthesis to integrate multi-source evidence, reduce inconsistencies, and improve overall faithfulness under complex queries.

### A.5.3. TIERING STRATEGY: MODULE'S MODEL CAPACITY

For **capacity tiering**, we operationalize the LOW/MID/HIGH budgets by *swapping the backbone LLM used inside the same module*, while keeping the module interface and inference behavior fixed. To control for prompt effects across tiers, we further *standardize the prompting* by using the same HIGH-tier prompt template for each module at all budget levels. In our implementation, tiers correspond to increasing model capacities drawn from widely used open-weight families: LOW instantiates modules with a small model (e.g., `Llama-3.2-3B-Instruct` and `qwen/qwen2.5-7b-instruct`), MID uses a medium model (e.g., `Llama-3.1-8B-Instruct` and `qwen/qwq-32b`), and HIGH uses a large model (e.g., `Llama-3.3-70B-Instruct` and `qwen3-next-80b-a3b-instruct`).

**Filter Module.** The filter module preserves an identical relevance scoring and selection procedure across tiers, and varies only the capacity of the scoring backbone. Given the query and a set of candidate memories, it assigns relevance scores and returns a ranked top-$k$ subset for downstream extraction. Under LOW/MID/HIGH, this scoring is executed by increasingly larger backbones, which improves semantic calibration and robustness under the same top-$k$ contract.

**Entity Module.** The entity module keeps the same extraction objective and output format across tiers, producing a standardized list of entity-centric relations from the filtered evidence. Capacity tiering affects only the model used to instantiate the extractor: larger backbones better recover implicit relations, while the emitted relation list remains interface-compatible with downstream fusion.

**Temporal Module.** The temporal module follows a fixed temporal extraction procedure for all tiers and outputs a standardized set of temporal facts/anchors. Capacity tiering changes only the underlying backbone used to infer and normalize temporally relevant constraints from the filtered memories, trading temporal fidelity and consistency against runtime cost without modifying the module contract.

**Topic Module.** The topic module maintains a fixed topic-relation extraction function and emits a standardized topic-cue list across all tiers. By varying only the backbone capacity, larger models can induce more stable topical structure and discourse transitions from the same evidence, while the interface and downstream consumption remain unchanged.

**Summary Module.** The summary module aggregates the upstream module outputs into a final compact memory representation under an identical summarization interface across tiers. Capacity tiering instantiates the summarizer with increasingly larger backbones, improving synthesis robustness and conflict handling under complex evidence, while keeping the final memory format and downstream expectations fixed.

## B. More Experimental Results

For LoCoMo (Maharana et al., 2024), we report the per-category performance in terms of F1-score, Judge score, and Cost. The results are computed separately for each category defined in the dataset, providing a fine-grained analysis of model performance across different long-context conversational scenarios. Following prior practice, we exclude Category 5 (adversarial) from our evaluation. To facilitate a clear comparison across model families, the results are reported in two separate tables, corresponding to LLaMA-based models and Qwen-based models, respectively. The detailed category-wise results on LoCoMo are summarized in Tables 7 and 8.

For LongMemEval (Wu et al., 2024), we present the category-level evaluation results, including F1-score, Judge score, and Cost. Each category is evaluated independently to reflect the model's ability to retain, retrieve, and reason over long-term contextual information under different memory-intensive settings. The results are organized into two separate tables for LLaMA-based and Qwen-based models, respectively, enabling a direct comparison between the two model families. The complete per-category results for LongMemEval are reported in Tables 9 and 10.

For HotpotQA (Yang et al., 2018), we provide the per-category performance measured by F1-score, Judge score, and Cost. The results are organized according to the predefined categories of the dataset, enabling a detailed examination of model behavior across different multi-hop reasoning types. For clarity, we report the results of LLaMA-based models and Qwen-based models in separate tables. The full category-wise results on HotpotQA are shown in Tables 11 and 12.

## C. Prompts

### C.1. Answer Prompt for the Locomo Dataset

```
Answer Prompt for the Locomo Dataset

Based on the above context, write an answer in the form of a short phrase for the
↪   following question. Answer with exact words from the context whenever possible.

Question: {} Short answer:
```

*Table 7.* **LoCoMo (LLaMA-based models)**. Per-category results in terms of **F1-score / Judge score** (%) and **Cost** ($).

| Method | Single-hop (F1/J) | Multi-hop (F1/J) | Temporal (F1/J) | Open-domain (F1/J) | Avg. (F1/J) | Cost ($) |
|---|---|---|---|---|---|---|
| **Baselines** | | | | | | |
| MemoryBank | 31.45/38.44 | 19.82/28.26 | 5.30/9.23 | 12.52/20.00 | 22.27/28.98 | 0.73 |
| ReadAgent | 28.67/36.56 | 25.65/42.75 | 5.55/4.62 | 17.06/32.50 | 22.48/31.05 | 0.57 |
| MemoryOS | 33.77/41.88 | 34.91/34.06 | 20.49/16.15 | 23.50/37.50 | 30.62/34.55 | 1.97 |
| A-MEM | 33.69/41.56 | 27.60/42.03 | 11.18/6.15 | 13.92/20.00 | 26.43/32.96 | 2.88 |
| Mem0 | 13.61/34.06 | 12.22/32.61 | 2.79/1.54 | 13.22/52.50 | 11.04/28.18 | 2.89 |
| LangMem | 25.90/29.68 | 28.22/36.23 | 6.14/6.15 | 25.67/25.00 | 22.31/25.96 | 0.48 |
| LightMem | 39.13/41.90 | 33.01/35.25 | 37.94/42.27 | 40.83/47.17 | 33.88/40.76 | 1.50 |
| **BudgetMem-IMP (COE)** | | | | | | |
| COE=0 | 43.76/57.81 | 34.01/42.75 | 36.88/38.46 | 21.09/55.00 | 38.75/50.32 | 1.80 |
| 0.05 | 43.61/55.94 | 32.98/43.48 | 37.93/40.77 | 23.13/55.00 | 38.80/50.00 | 1.75 |
| 0.1 | 46.55/59.38 | 34.37/44.93 | 44.57/43.08 | 26.81/55.00 | 42.21/52.55 | 1.26 |
| 0.3 | 23.72/31.25 | 23.99/32.61 | 14.72/12.31 | 10.48/30.00 | 21.07/27.55 | 0.45 |
| 0.5 | 25.19/31.56 | 25.47/36.23 | 16.64/13.85 | 9.52/27.50 | 22.48/28.66 | 0.48 |
| 0.7 | 28.76/36.25 | 26.51/37.68 | 18.09/14.62 | 9.37/32.50 | 24.82/31.85 | 0.48 |
| 0.9 | 25.77/32.81 | 24.73/36.23 | 22.27/19.23 | 19.05/37.50 | 24.39/31.05 | 0.48 |
| **BudgetMem-REA (COE)** | | | | | | |
| COE=0 | 44.95/57.19 | 30.32/47.10 | 44.48/44.62 | 33.77/55.00 | 40.92/52.23 | 2.90 |
| 0.05 | **47.55**/61.88 | 32.83/49.28 | 37.07/39.23 | 32.16/60.00 | 41.16/54.30 | 2.65 |
| 0.1 | 46.44/60.00 | 30.65/43.48 | 44.36/44.62 | 37.28/65.00 | 41.95/53.50 | 2.72 |
| 0.3 | 42.34/57.19 | 30.16/42.03 | 40.64/40.00 | 25.15/50.00 | 38.22/49.84 | 2.33 |
| 0.5 | 41.61/57.50 | 24.97/47.10 | 40.65/43.08 | 22.02/50.00 | 36.51/51.75 | 2.34 |
| 0.7 | 40.64/51.88 | 32.51/47.10 | 41.72/**49.23** | 24.19/47.50 | 38.03/50.00 | 2.33 |
| 0.9 | 40.55/53.12 | 31.82/44.20 | 35.03/39.23 | 21.46/47.50 | 36.27/47.93 | 2.32 |
| **BudgetMem-CAP (COE)** | | | | | | |
| COE=0 | **47.55**/60.62 | 34.15/50.72 | **45.19**/43.08 | 30.76/57.50 | **43.05**/54.62 | 2.40 |
| 0.05 | 45.73/59.06 | 32.20/45.65 | 44.32/46.15 | 37.64/67.50 | 41.95/53.98 | 2.48 |
| 0.1 | 45.76/**64.06** | 33.26/**52.17** | 44.93/43.85 | 39.01/**70.00** | 42.41/**57.64** | 2.48 |
| 0.3 | 32.35/60.31 | **40.53**/46.38 | 34.46/38.46 | **45.32**/60.00 | 40.79/52.71 | 2.00 |
| 0.5 | 40.52/46.88 | 27.53/39.86 | 30.63/24.62 | 22.25/47.50 | 34.45/40.76 | 0.58 |
| 0.7 | 29.73/39.37 | 24.32/30.43 | 28.80/13.85 | 22.80/50.00 | 27.91/32.80 | **0.28** |
| 0.9 | 26.78/34.38 | 21.44/28.99 | 25.11/12.31 | 18.87/47.50 | 24.76/29.46 | **0.28** |

## C.2. Answer Prompt for the HotpotQA Dataset

Answer Prompt for the HotpotQA Dataset

```
Based on the following context, answer the question. The question may require
↪   reasoning across multiple pieces of information.

{context}

Question: {question}

Instructions:
- Read the context carefully and identify relevant information
- If the answer can be found in the context, provide a short, precise answer
- Output your answer within <answer></answer> tags

<answer>your answer here</answer>
```

*Table 8.* **LoCoMo (Qwen-based models)**. Per-category results (**F1/Judge, %**) and cost ($).

| Method | Single-hop (F1/J) | Multi-hop (F1/J) | Temporal (F1/J) | Open-domain (F1/J) | **Avg.** (F1/J) | **Cost** ($) |
|---|---|---|---|---|---|---|
| **Baselines** | | | | | | |
| MemoryBank | 30.50/44.37 | 22.80/36.23 | 6.44/6.15 | 25.88/45.00 | 23.53/34.71 | 0.25 |
| MemoryOS | 42.05/45.65 | 26.59/15.38 | 42.98/**60.00** | **35.23**/42.81 | 35.43/38.85 | 0.75 |
| ReadAgent | 29.04/36.88 | 23.71/40.58 | 3.93/3.85 | 25.56/45.00 | 22.45/31.37 | 0.24 |
| A-MEM | 32.02/45.31 | 26.66/46.38 | 17.81/9.23 | 28.12/52.50 | 27.65/38.54 | 2.88 |
| LangMem | 24.49/27.18 | 24.55/30.43 | 4.99/6.15 | 31.21/25.00 | 20.89/23.40 | 0.14 |
| Mem0 | 11.52/28.77 | 15.25/31.08 | 3.54/3.03 | 12.19/46.15 | 10.77/25.32 | 1.15 |
| LightMem | 39.45/52.81 | 23.08/38.41 | 23.93/23.08 | 27.77/42.50 | 32.85/42.83 | 0.70 |
| **BudgetMem-IMP (COE)** | | | | | | |
| COE=0 | 46.52/62.90 | 28.91/46.09 | 38.39/40.98 | 30.63/55.88 | 40.14/54.38 | 0.80 |
| 0.05 | 45.58/59.29 | 30.12/43.94 | 44.09/45.24 | 24.22/52.63 | 40.58/52.63 | 0.52 |
| 0.1 | **46.92**/61.08 | 31.00/44.78 | 43.87/39.68 | 31.40/**63.89** | **41.89**/53.27 | 0.46 |
| 0.3 | 30.09/35.94 | 26.52/36.23 | 21.25/16.15 | 25.04/55.00 | 27.15/33.12 | **0.07** |
| 0.5 | 27.79/34.06 | 25.12/34.06 | 21.37/13.08 | 23.74/55.00 | 25.62/31.05 | 0.10 |
| 0.7 | 25.98/32.81 | 26.14/35.51 | 26.23/19.23 | 25.17/52.50 | 26.01/31.85 | 0.09 |
| 0.9 | 26.05/32.19 | 25.48/32.61 | 22.62/16.92 | 24.87/55.00 | 25.14/30.57 | 0.09 |
| **BudgetMem-REA (COE)** | | | | | | |
| COE=0 | 45.47/63.44 | 32.41/45.65 | 37.57/34.62 | 33.31/60.00 | 40.19/53.34 | 1.11 |
| 0.05 | 44.49/60.94 | 34.63/**50.72** | 40.52/36.92 | 30.39/55.00 | 41.00/53.34 | 0.62 |
| 0.1 | 45.17/18.24 | 33.14/6.07 | 40.63/11.93 | 28.46/5.67 | 40.52/55.25 | 0.61 |
| 0.3 | 44.20/62.81 | 33.05/46.38 | 41.70/38.46 | 33.41/60.00 | 40.55/53.98 | 0.61 |
| 0.5 | 43.78/63.44 | 34.60/47.83 | **46.00**/44.62 | 26.94/60.00 | 41.15/**55.89** | 0.61 |
| 0.7 | 44.54/64.06 | 32.25/45.65 | 43.29/41.54 | 28.80/47.50 | 40.58/54.30 | 0.61 |
| 0.9 | 46.47/63.44 | **37.43**/50.00 | 38.96/40.00 | 26.84/55.00 | 41.68/55.10 | 0.62 |
| **BudgetMem-CAP (COE)** | | | | | | |
| COE=0 | 46.18/63.12 | 33.14/46.38 | 40.37/34.62 | 32.19/57.50 | 41.22/53.18 | 0.61 |
| 0.05 | 44.82/62.50 | 35.96/47.10 | 42.45/39.23 | 28.65/55.00 | 41.35/53.82 | 0.61 |
| 0.1 | 45.64/**66.25** | 32.73/44.20 | 43.34/40.77 | 29.54/60.00 | 41.30/55.73 | 0.61 |
| 0.3 | 46.15/64.38 | 32.92/45.65 | 43.50/44.62 | 29.97/55.00 | 41.66/55.57 | 0.61 |
| 0.5 | 45.88/61.25 | 34.27/45.65 | 40.92/37.69 | 28.81/50.00 | 41.21/52.23 | 0.56 |
| 0.7 | 44.79/65.00 | 36.50/46.38 | 42.38/39.23 | 32.66/57.50 | 41.65/55.10 | 0.56 |
| 0.9 | 45.21/62.19 | 31.86/43.48 | 40.73/35.38 | 34.88/57.50 | 40.69/52.23 | 0.55 |

## C.3. Answer Prompt for the LongMemEval Dataset

> Answer Prompt for the LongMemEval Dataset
>
> ```
> I will give you several history chats between you and a user.
>
> Please answer the question based on the relevant chat history.
>
> History Chats:{}
> Current Date: {}
> Question: {}
> Short Answer:
> ```

*Table 9.* **LongMemEval (LLaMA-based models)**. Per-category results (**F1/Judge, %**) and cost ($).

| Method | SS-User (F1/J) | Multi-Session (F1/J) | SS-Pref (F1/J) | Temporal (F1/J) | Know.-Update (F1/J) | SS-Assist. (F1/J) | **Avg.** (F1/J) | **Cost** ($) |
|---|---|---|---|---|---|---|---|---|
| **Baselines** | | | | | | | | |
| MemoryBank | 22.64/17.86 | 35.87/42.86 | 76.33/79.17 | 11.10/16.67 | 8.11/19.23 | 24.75/43.33 | 26.74/32.67 | 3.94 |
| ReadAgent | 16.91/12.50 | 60.71/67.86 | 14.52/8.33 | 22.35/33.33 | 2.77/15.38 | 26.14/53.33 | 20.75/27.72 | 13.68 |
| MemoryOS | 13.03/24.14 | 22.88/75.00 | 16.83/42.86 | 15.71/33.33 | 2.11/6.52 | 15.77/46.43 | 12.97/33.50 | 38.83 |
| A-MEM | 22.05/17.86 | 67.86/64.29 | **76.89**/83.33 | 14.60/8.33 | 12.44/15.38 | 23.93/26.67 | 21.74/33.17 | 80.02 |
| Mem0 | 22.31/25.00 | 73.48/82.14 | 19.47/41.67 | 19.89/41.67 | 4.78/23.08 | **44.46/70.00** | 27.70/42.08 | 13.57 |
| LangMem | 15.92/10.34 | 18.75/35.71 | 13.76/25.00 | 13.96/25.00 | 3.05/8.70 | 9.23/14.29 | 12.00/17.00 | 16.60 |
| LightMem | **74.31/82.14** | 8.10/50.00 | 23.12/58.33 | 19.79/28.57 | **36.99/70.00** | 16.88/20.83 | 26.74/48.51 | 5.28 |
| **BudgetMem-IMP (COE)** | | | | | | | | |
| COE=0 | 41.52/55.17 | 67.40/71.43 | 38.26/89.29 | 17.99/50.00 | 28.98/41.30 | 20.67/35.71 | 37.47/56.00 | 0.71 |
| 0.05 | 22.23/41.38 | 59.74/75.00 | 25.04/42.86 | 23.49/**66.67** | 6.00/8.70 | 19.44/42.86 | 23.83/40.50 | 0.54 |
| 0.1 | 20.80/43.10 | 64.14/67.86 | 21.98/42.86 | 22.41/**66.67** | 5.35/8.70 | 14.26/39.29 | 22.66/39.50 | 0.58 |
| 0.3 | 25.07/37.93 | 61.35/75.00 | 25.23/42.86 | 23.38/58.33 | 5.28/10.87 | 21.21/46.43 | 24.98/40.00 | 0.58 |
| 0.5 | 18.35/46.55 | 64.11/71.43 | 25.03/42.86 | 20.59/50.00 | 5.31/8.70 | 21.19/50.00 | 23.22/41.50 | 0.58 |
| 0.7 | 17.78/43.10 | 59.95/67.86 | 24.45/42.86 | 22.95/58.83 | 6.03/8.70 | 19.53/53.57 | 22.47/41.00 | 0.58 |
| 0.9 | 23.29/39.66 | 58.05/67.86 | 25.87/42.86 | **24.92**/58.33 | 5.98/8.70 | 18.99/46.43 | 24.03/39.00 | 0.58 |
| **BudgetMem-REA (COE)** | | | | | | | | |
| COE=0 | 40.51/53.45 | 73.21/82.14 | 45.28/89.29 | 22.77/50.00 | 31.39/36.96 | 25.75/50.00 | 40.53/58.00 | 0.67 |
| 0.05 | 38.49/44.83 | 77.11/82.14 | 40.46/**92.86** | 24.45/58.33 | 35.54/41.30 | 28.79/42.86 | 41.29/56.50 | 0.66 |
| 0.1 | 42.38/48.28 | 76.79/82.14 | 44.55/89.29 | 20.55/41.67 | 23.65/36.96 | 17.10/42.86 | 38.34/55.00 | 0.62 |
| 0.3 | 34.90/51.72 | 77.11/**85.71** | 30.37/75.00 | 17.52/25.00 | 25.87/39.13 | 24.75/46.43 | 35.63/54.50 | 0.67 |
| 0.5 | 35.84/46.55 | 77.11/**85.71** | 42.88/89.29 | 18.81/41.67 | 32.30/30.43 | 30.46/42.86 | 40.01/53.50 | 0.68 |
| 0.7 | 36.48/48.28 | 77.98/82.14 | 40.64/89.29 | 20.70/50.00 | 32.91/39.13 | 26.24/39.29 | 39.67/55.50 | 0.68 |
| 0.9 | 33.66/44.83 | 76.79/85.71 | 34.55/**92.86** | 17.94/33.33 | 20.66/32.61 | 27.95/46.43 | 35.09/54.00 | 0.61 |
| **BudgetMem-CAP (COE)** | | | | | | | | |
| COE=0 | 36.78/50.00 | 75.60/82.14 | 39.73/**92.86** | 19.46/58.33 | 34.38/43.48 | 31.08/57.14 | 40.24/**60.50** | 0.80 |
| 0.05 | 36.19/53.45 | 72.01/82.14 | 36.80/82.14 | 21.65/33.33 | 30.71/47.83 | 29.31/39.29 | 38.19/57.00 | 0.63 |
| 0.1 | 35.16/41.38 | 69.64/78.57 | 42.49/89.29 | 19.56/50.00 | 24.54/30.43 | 7.84/21.43 | 33.81/48.50 | 0.13 |
| 0.3 | 39.59/53.45 | **80.36/85.71** | 39.59/89.29 | 18.44/16.67 | 32.31/41.30 | 38.80/46.43 | **42.25**/57.00 | **0.10** |
| 0.5 | 29.28/34.48 | 60.97/67.86 | 29.14/67.86 | 17.58/33.33 | 14.07/17.39 | 11.36/25.00 | 26.98/38.50 | **0.10** |
| 0.7 | 25.30/31.03 | 59.88/67.86 | 36.48/71.43 | 18.28/41.67 | 14.67/21.74 | 10.62/17.86 | 26.79/38.50 | **0.10** |
| 0.9 | 33.77/34.48 | 60.97/60.71 | 23.56/53.57 | 18.72/33.33 | 14.98/17.39 | 17.29/32.14 | 28.62/36.50 | **0.10** |

*Table 10.* **LongMemEval (Qwen-based models)**. Per-category results (**F1/Judge, %**) and cost ($).

| Method | SS-User (F1/J) | Multi-Session (F1/J) | SS-Pref (F1/J) | Temporal (F1/J) | Know.-Update (F1/J) | SS-Assist. (F1/J) | **Avg.** (F1/J) | **Cost** ($) |
|---|---|---|---|---|---|---|---|---|
| **Baselines** | | | | | | | | |
| MemoryBank | 10.41/12.50 | 5.08/14.29 | 15.39/33.33 | 14.54/**75.00** | 9.36/23.08 | 12.58/56.67 | 10.56/28.22 | 3.45 |
| MemoryOS | 13.45/27.59 | 25.24/75.00 | 17.59/42.86 | 13.47/25.00 | 2.57/4.35 | 14.64/42.86 | 13.35/33.00 | 15.84 |
| ReadAgent | 13.65/14.29 | 41.39/64.29 | 8.83/8.33 | 11.93/58.33 | 2.75/15.38 | 12.49/46.67 | 27.72/20.75 | 4.97 |
| A-MEM | 14.66/7.14 | 46.81/53.57 | **65.14**/79.17 | 15.18/16.67 | 3.60/19.23 | 13.62/33.33 | 10.82/31.19 | 21.00 |
| LangMem | 16.68/10.34 | 10.43/25.00 | 13.58/28.57 | 13.64/33.33 | 4.14/0.00 | 7.44/10.71 | 11.01/14.00 | 3.99 |
| Mem0 | 17.93/14.29 | **78.84**/85.71 | 21.06/45.83 | **20.66**/41.67 | 4.95/18.31 | 32.34/53.33 | 25.71/36.14 | 4.96 |
| LightMem | **65.70/85.71** | 17.84/50.00 | 13.81/66.66 | 18.49/12.50 | **36.26/80.00** | 29.16/22.50 | 27.70/47.52 | 3.39 |
| **BudgetMem-IMP (COE)** | | | | | | | | |
| COE=0 | 32.47/32.76 | 43.92/85.71 | 42.07/92.86 | 17.08/66.67 | 11.49/26.09 | 29.01/53.57 | 29.18/52.00 | 0.30 |
| 0.05 | 29.02/44.83 | 53.92/85.71 | 31.27/92.86 | 20.27/50.00 | 21.50/34.78 | 21.62/46.43 | 29.53/55.50 | 0.41 |
| 0.1 | 12.73/37.93 | 33.08/85.71 | 23.06/60.71 | 15.20/66.67 | 6.74/17.39 | 6.32/46.43 | 14.90/46.00 | **0.13** |
| 0.3 | 12.54/49.29 | 25.26/75.00 | 9.20/57.14 | 15.64/58.33 | 2.20/8.70 | 13.45/42.86 | 11.41/44.00 | 0.15 |
| 0.5 | 12.16/43.10 | 24.85/78.57 | 11.79/46.43 | 13.42/50.00 | 2.19/10.87 | 13.40/46.43 | 11.84/42.00 | 0.15 |
| 0.7 | 11.07/39.66 | 22.57/75.00 | 8.40/53.57 | 15.33/**75.00** | 2.27/8.70 | 13.76/42.86 | 10.91/42.00 | 0.15 |
| 0.9 | 11.39/44.83 | 22.44/78.57 | 12.63/64.29 | 14.38/58.33 | 2.15/8.70 | 14.02/42.86 | 11.39/44.50 | 0.15 |
| **BudgetMem-REA (COE)** | | | | | | | | |
| COE=0 | 40.99/48.28 | 63.24/85.71 | 38.56/89.29 | 17.38/66.67 | 20.89/43.48 | 27.56/46.43 | **35.84/59.00** | 0.26 |
| 0.05 | 40.67/41.38 | 49.64/85.71 | 43.22/82.14 | 16.30/66.67 | 13.64/39.13 | 26.16/42.86 | 32.57/54.50 | 0.17 |
| 0.1 | 35.64/34.48 | 49.31/85.71 | 37.05/75.00 | 16.03/58.33 | 20.87/47.83 | 31.23/50.00 | 32.56/54.00 | 0.17 |
| 0.3 | 37.32/37.93 | 47.10/85.71 | 43.43/85.71 | 16.77/66.67 | 14.92/39.13 | 26.22/53.57 | 31.60/55.50 | 0.17 |
| 0.5 | 38.58/41.38 | 57.88/82.14 | 47.13/85.71 | 16.96/66.67 | 16.52/43.48 | **35.06/60.71** | 35.62/58.00 | 0.17 |
| 0.7 | 33.56/37.93 | 53.60/85.71 | 38.49/89.29 | 16.60/41.67 | 21.91/47.83 | 24.12/46.43 | 32.04/55.50 | 0.17 |
| 0.9 | 40.91/44.83 | 54.31/85.71 | 34.74/85.71 | 15.68/66.67 | 24.34/43.48 | 26.05/50.00 | 34.52/58.00 | 0.17 |
| **BudgetMem-CAP (COE)** | | | | | | | | |
| COE=0 | 30.47/34.48 | 49.77/85.71 | 43.91/92.86 | 16.55/58.33 | 16.55/47.83 | 22.86/46.43 | 32.01/56.00 | 0.17 |
| 0.05 | 38.31/41.38 | 58.60/85.71 | 40.21/82.14 | 17.89/66.67 | 20.49/47.83 | 26.98/46.43 | 34.51/57.00 | 0.17 |
| 0.1 | 31.56/34.48 | 48.16/85.71 | 44.08/92.86 | 16.16/66.67 | 23.01/47.83 | 27.61/53.57 | 32.19/57.50 | 0.17 |
| 0.3 | 36.57/41.38 | 54.31/85.71 | 46.93/85.71 | 17.38/66.67 | 17.30/43.48 | 27.76/46.43 | 33.69/56.50 | 0.17 |
| 0.5 | 29.30/29.31 | 51.08/**92.86** | 43.26/**100.00** | 14.58/33.33 | 12.96/28.26 | 16.15/46.43 | 27.82/50.50 | 0.22 |
| 0.7 | 34.46/36.21 | 49.14/85.71 | 42.23/92.86 | 12.89/50.00 | 14.38/32.61 | 11.06/42.86 | 28.42/52.00 | 0.17 |
| 0.9 | 33.79/34.48 | 56.25/89.29 | 44.92/92.86 | 13.38/50.00 | 16.89/32.61 | 14.54/46.43 | 30.68/52.50 | 0.22 |

*Table 11.* **HotpotQA (LLaMA-based models)**. Overall results (**F1/Judge, %**) and cost ($).

| Method | **Overall** (F1/J) | **Cost** ($) |
|---|---|---|
| **Baselines** | | |
| MemoryBank | 22.25/23.75 | 7.75 |
| ReadAgent | 15.33/30.08 | 4.19 |
| MemoryOS | 34.50/43.36 | 13.32 |
| A-MEM | 43.25/54.69 | 26.74 |
| Mem0 | 28.03/36.72 | 4.30 |
| LangMem | 22.78/22.66 | 10.95 |
| LightMem | 45.73/58.37 | 10.10 |
| **BudgetMem-IMP (COE)** | | |
| COE=0 | 49.31/65.77 | 1.35 |
| 0.05 | 51.40/63.28 | 1.23 |
| 0.1 | 48.49/60.94 | 1.15 |
| 0.3 | 40.95/45.70 | 1.14 |
| 0.5 | 37.40/44.14 | 1.37 |
| 0.7 | 37.52/43.36 | 1.37 |
| 0.9 | 39.01/44.53 | 1.37 |
| **BudgetMem-REA (COE)** | | |
| COE=0 | 51.12/61.93 | 0.99 |
| 0.05 | 51.23/63.55 | 0.95 |
| 0.1 | 53.93/66.36 | 0.77 |
| 0.3 | **57.17/68.40** | 0.79 |
| 0.5 | 56.86/66.07 | 0.82 |
| 0.7 | 56.66/**68.40** | 0.77 |
| 0.9 | 55.04/66.20 | 0.80 |
| **BudgetMem-CAP (COE)** | | |
| COE=0 | 53.87/64.85 | 0.93 |
| 0.05 | 38.62/50.00 | 0.84 |
| 0.1 | 31.71/36.27 | **0.10** |
| 0.3 | 24.65/32.57 | 0.11 |
| 0.5 | 22.16/28.90 | 0.11 |
| 0.7 | 30.92/37.62 | **0.10** |
| 0.9 | 35.29/36.57 | **0.10** |

*Table 12.* **HotpotQA (Qwen-based models)**. Overall results (**F1/Judge, %**) and cost ($).

| Method | Overall (F1/J) | Cost ($) |
|---|---|---|
| **Baselines** | | |
| MemoryBank | 18.64/31.25 | 1.79 |
| MemoryOS | 41.21/53.52 | 11.68 |
| ReadAgent | 18.05/25.78 | 1.75 |
| A-MEM | 40.54/50.39 | 8.34 |
| LangMem | 20.77/21.09 | 4.32 |
| Mem0 | 24.72/37.89 | 2.02 |
| LightMem | 41.29/55.42 | 8.56 |
| **BudgetMem-IMP (COE)** | | |
| COE=0 | 46.67/57.42 | 0.63 |
| 0.05 | 32.33/42.58 | 0.26 |
| 0.1 | 34.01/42.97 | 0.26 |
| 0.3 | 33.51/44.14 | 0.30 |
| 0.5 | 30.17/41.02 | 0.30 |
| 0.7 | 31.53/42.97 | 0.30 |
| 0.9 | 32.26/42.58 | 0.30 |
| **BudgetMem-REA (COE)** | | |
| COE=0 | 57.67/70.83 | 0.17 |
| 0.05 | 50.74/58.50 | 0.32 |
| 0.1 | 60.26/69.89 | 0.18 |
| 0.3 | 57.04/68.75 | 0.19 |
| 0.5 | 59.28/69.79 | 0.19 |
| 0.7 | 58.79/70.00 | 0.19 |
| 0.9 | 59.96/71.43 | **0.12** |
| **BudgetMem-CAP (COE)** | | |
| COE=0 | 58.70/72.08 | 0.22 |
| 0.05 | 59.57/71.91 | 0.18 |
| 0.1 | 60.03/71.51 | 0.19 |
| 0.3 | 59.61/71.98 | 0.18 |
| 0.5 | **64.24/74.75** | 0.20 |
| 0.7 | 58.63/69.41 | 0.17 |
| 0.9 | 59.90/71.05 | 0.15 |

## C.4. LLM JUDGE GENERAL PROMPT

---
**LLM JUDGE GENERAL PROMPT**

You are an expert judge evaluating the quality of an answer for a QA task.
Your goal is to determine whether the model's answer correctly and sufficiently
answers the given question.

Read the following information carefully:

[Question]
{question}

[Ground Truth Answers]
{ground_truth}

[Model Answer]
{model_answer}

Your evaluation criteria:
1. Correctness:
    - Is the model answer factually consistent with ANY of the correct answers?
    - Does it avoid contradictions or introducing false information?

2. Relevance:
    - Does the answer address the question directly without unnecessary content?

3. Completeness:
    - Does the answer include all essential information needed to fully answer the
    ↪  question?
    - Partial answers are allowed but should receive lower scores.

Scoring Rules:
- Score = 1.0 if the answer is fully correct.
- Score = 0.5 if the answer is partially correct but incomplete or slightly
↪  inaccurate.
- Score = 0.0 if the answer is incorrect, irrelevant, or contradicts the ground
↪  truth.

Output Format (STRICT):
Return your output as a JSON dictionary with two fields:
{{
    "explanation": "<brief explanation of your reasoning>",
    "score": <0.0 | 0.5 | 1.0>
}}

Be concise and objective. Do not include anything outside the JSON.

---

## C.5. FILTER MODULE PROMPT

FILTER MODULE PROMPT (Low)

```
**Role:** You are a relevance scoring system.
**Task:** Given a query and an ordered list of memories, output a score for
↪   each memory indicating how directly and usefully it helps answer the
↪   query.

**Scoring Rules:**
- 10: directly answers the query or provides essential constraints/info to
↪   answer it
- 7{9: clearly relevant and helpful, but not fully sufficient alone
- 4{6: partially relevant; some usefulness but missing key connection
- 1{3: weak/tangential relevance; mostly not useful
- 0: completely irrelevant

**Input Format:**
The input consists of:
1. A `<query>` section containing the user's question.
2. A `<memories>` section containing individual `<memory>` elements.
Each memory is formatted as:
```
<memory index="N" [date_time="..." session_id="..." dia_id="..."]>
memory content text
</memory>
```
Where:
- `index` is the memory's position in the list.
- `date_time`, `session_id`, `dia_id` are optional metadata attributes.
- The text between the tags is the memory content to be scored.

**Output Format:**
Your entire response must be **only** the following line:
`<answer>[s0, s1, s2, ...]</answer>`
- The array must contain exactly one integer score per memory, in the same
↪   order as the input indices.
- If there are no memories, output: `<answer>[]</answer>`

**Input:**
<query>{query}</query>
<memories>{memories_text}</memories>
```

## Filter Module Prompt (Mid)

**Role:** You are a relevance scoring system.
**Task:** Given a query and an ordered list of memories, output a score for
↪  each memory indicating how directly and usefully it helps answer the
↪  query.

**Scoring Rules:**
- 10: directly answers the query or provides essential constraints/info to
↪  answer it
- 7{9: clearly relevant and helpful, but not fully sufficient alone
- 4{6: partially relevant; some usefulness but missing key connection
- 1{3: weak/tangential relevance; mostly not useful
- 0: completely irrelevant

**Input Format:**
The input consists of:
1. A `<query>` section containing the user's question.
2. A `<memories>` section containing individual `<memory>` elements.
Each memory is formatted as:
```
<memory index="N" [date_time="..." session_id="..." dia_id="..."]>
memory content text
</memory>
```
Where:
- `index` is the memory's position in the list.
- `date_time`, `session_id`, `dia_id` are optional metadata attributes.
- The text between the tags is the memory content to be scored.

**Output Format:**
Your entire response must be **only** the following line:
`<answer>[s0, s1, s2, ...]</answer>`
- The array must contain exactly one integer score per memory, in the same
↪  order as the input indices.
- If there are no memories, output: `<answer>[]</answer>`

Let's think step by step to score the relevance of each memory to the query.

**Input:**
<query>{query}</query>
<memories>{memories_text}</memories>

FILTER MODULE PROMPT (High)

**Role:** You are a relevance scoring system.
**Task:** Given a query and an ordered list of memories, output a score for
↪   each memory indicating how directly and usefully it helps answer the
↪   query.

**Scoring Rules:**
− 9{10: directly answers the query or contains essential key facts
− 6{8: clearly helpful and strongly related to the query
− 3{5: somewhat related or only partially helpful
− 1{2: very weakly related (e.g., superficial keyword overlap only)
− 0: completely irrelevant or not useful for the query

**Input Format:**
The input consists of:
1. A `<query>` section containing the user's question.
2. A `<memories>` section containing individual `<memory>` elements.
Each memory is formatted as:
```

<memory index="N" [date_time="..." session_id="..." dia_id="..."]>
memory content text
</memory>
```

Where:
− `index` is the memory's position in the list.
− `date_time`, `session_id`, `dia_id` are optional metadata attributes.
− The text between the tags is the memory content to be scored.

To complete the task systematically, please follow the steps reasoning
↪   framework outlined below:

**Reasoning Steps:**
**PLAN:**
− Analyze the query's intent and define relevance criteria.
− Identify key scoring dimensions (e.g., topical alignment, factual
↪   support).

**ACT:**
− Evaluate each memory against the criteria.
− For each memory, briefly justify the provisional score based on its
↪   content.

**REFLECT:**
− Review all scores for consistency and calibration.
− Make adjustments if needed to ensure fairness and coherence.

**Output Format:**
First write your reasoning following the PLAN → ACT → REFLECT structure.
Then, your final response must be **only** the following line:
`<answer>[s0, s1, s2, ...]</answer>`
− The array must contain exactly one integer score per memory, in the same
↪   order as the input indices.
− If there are no memories, output: `<answer>[]</answer>`

**Input:**
<query>{query}</query>
<memories>{memories_text}</memories>

## C.6. ENTITY MODULE RELATION PROMPT

---

**ENTITY MODULE RELATION PROMPT (Low)**

```
**Role:** You are a specialist in entity semantic relationship extraction.
**Task:** From the provided memories, extract **only** concrete
↪    relationships between entities that are central to answering the given
↪    query.

**Extraction Criteria:**
- Focus on factual connections where the entities and their relationship
↪    directly help answer the query.
- Ignore background entities, anecdotes, or side topics.
- Each extracted relationship should be a concise, useful fact.

**Input Format:**
The `<memories>` section contains a list of `<memory>` tags.
Each memory is formatted as:
```
<memory [date_time="..." session_id="..." dia_id="..."]>
memory content text
</memory>
```
Where:
- `date_time`, `session_id`, `dia_id` are optional metadata attributes.
- The text within the tags is the content to analyze for relationships.

**Output Format:**
Your entire response must be a JSON array of relationship strings, formatted
↪    as follows:
`<answer>["relationship1", "relationship2", ...]</answer>`
Each relationship string must be in the format: `"EntityA - relation -
↪    EntityB (optional context)"`
If no relevant relationships are found, output: `<answer>[]</answer>`

**Input:**
<query>{query}</query>
<memories>{memories_text}</memories>
```

ENTITY MODULE RELATION PROMPT (Mid)

**Role:** You are a specialist in entity semantic relationship extraction.
**Task:** From the provided memories, extract **only** concrete
↪   relationships between entities that are central to answering the given
↪   query.

**Extraction Criteria:**
– Focus on factual connections where the entities and their relationship
↪   directly help answer the query.
– Ignore background entities, anecdotes, or side topics.
– Each extracted relationship should be a concise, useful fact.

**Input Format:**
The `<memories>` section contains a list of `<memory>` tags.
Each memory is formatted as:
```
<memory [date_time="..." session_id="..." dia_id="..."]>
memory content text
</memory>
```
Where:
– `date_time`, `session_id`, `dia_id` are optional metadata attributes.
– The text within the tags is the content to analyze for relationships.

**Output Format:**
Your entire response must be a JSON array of relationship strings, formatted
↪   as follows:
`<answer>["relationship1", "relationship2", ...]</answer>`
Each relationship string must be in the format: `"EntityA – relation –
↪   EntityB (optional context)"`
If no relevant relationships are found, output: `<answer>[]</answer>`

Let's think step by step to extract the relationships between entities.

**Input:**
<query>{query}</query>
<memories>{memories_text}</memories>

ENTITY MODULE RELATION PROMPT (High)

**Role:** You are a specialist in entity semantic relationship extraction.
**Task:** From the provided memories, extract **only** concrete
↪   relationships between entities that are central to answering the given
↪   query.

**Extraction Criteria:**
– Focus on factual connections where the entities and their relationship
↪   directly help answer the query.
– Ignore background entities, anecdotes, or side topics.
– Each extracted relationship should be a concise, useful fact.

**Input Format:**
The `<memories>` section contains a list of `<memory>` tags.
Each memory is formatted as:
```
<memory [date_time="..." session_id="..." dia_id="..."]>
memory content text
</memory>
```
Where:
– `date_time`, `session_id`, `dia_id` are optional metadata attributes.
– The text within the tags is the content to analyze for relationships.

To complete the task systematically, please follow the steps reasoning
↪   framework outlined below:

**Reasoning Steps:**
**PLAN:**
– Analyze the query to determine its precise intent.
– Identify which entity types and relationship types are essential for
↪   answering it.

**ACT:**
– From the memories, extract candidate relations that involve only
↪   query-relevant entities.
– Format each relation as a concise, factual statement.

**REFLECT:**
– **CHECK:** Verify each extracted relation:
  – Does it directly help answer the query?
  – Is it clearly supported by the memory content?
– **REGENERATE (if needed):** If important query-relevant relations are
↪   missing, re-extract or rewrite them to better align with the query
↪   intent.

**Output Format:**
First write your reasoning following the PLAN → ACT → REFLECT structure.
Then, your final response must be **only** the following line:
`<answer>["relationship1", "relationship2", ...]</answer>`
Each relationship string must be in the format: `"EntityA – relation –
↪   EntityB (optional context)"`
If no relevant relationships are found, output: `<answer>[]</answer>`

**Input:**
<query>{query}</query>
<memories>{memories_text}</memories>

## C.7. TEMPORAL MODULE RELATION PROMPT

TEMPORAL MODULE RELATION PROMPT (Low)

**Role:** You are a specialist in temporal information extraction.
**Task:** From the provided memories, extract **only** specific temporal
↪ facts that are necessary for answering the given query.

**Extraction Criteria:**
- Focus on precise temporal information that directly constrains or
↪ clarifies the answer.
- Ignore vague, irrelevant, or background time references.
- Each extracted item must represent a clear, factual temporal statement.

**Input Format:**
The `<memories>` section contains a list of `<memory>` tags.
Each memory is formatted as:
```
<memory [date_time="..." session_id="..." dia_id="..."]>
memory content text
</memory>
```
Where:
- `date_time`, `session_id`, `dia_id` are optional metadata attributes.
- The text within the tags is the content to analyze for temporal
↪ information.

**Output Format:**
Your entire response must be a JSON array of temporal fact strings,
↪ formatted as follows:
`<answer>["temporal_fact1", "temporal_fact2", ...]</answer>`
Each string should clearly express a temporal fact (e.g., "Event occurred in
↪ March 2023", "Process lasted 3 days", "A happened before B").
If no relevant temporal information is found, output: `<answer>[]</answer>`

**Input:**
<query>{query}</query>
<memories>{memories_text}</memories>

TEMPORAL MODULE RELATION PROMPT (Mid)

**Role:** You are a specialist in temporal information extraction.
**Task:** From the provided memories, extract **only** specific temporal
↪  facts that are necessary for answering the given query.

**Extraction Criteria:**
– Focus on precise temporal information that directly constrains or
↪  clarifies the answer.
– Ignore vague, irrelevant, or background time references.
– Each extracted item must represent a clear, factual temporal statement.

**Input Format:**
The `<memories>` section contains a list of `<memory>` tags.
Each memory is formatted as:
```
<memory [date_time="..." session_id="..." dia_id="..."]>
memory content text
</memory>
```
Where:
– `date_time`, `session_id`, `dia_id` are optional metadata attributes.
– The text within the tags is the content to analyze for temporal
↪  information.

**Output Format:**
Your entire response must be a JSON array of temporal fact strings,
↪  formatted as follows:
`<answer>["temporal_fact1", "temporal_fact2", ...]</answer>`
Each string should clearly express a temporal fact (e.g., "Event occurred in
↪  March 2023", "Process lasted 3 days", "A happened before B").
If no relevant temporal information is found, output: `<answer>[]</answer>`

Let's think step by step to extract the temporal information.

**Input:**
<query>{query}</query>
<memories>{memories_text}</memories>

TEMPORAL MODULE RELATION PROMPT (High)

**Role:** You are a specialist in temporal information extraction.
**Task:** From the provided memories, extract **only** specific temporal
↪  facts that are necessary for answering the given query.

**Extraction Criteria:**
– Focus on precise temporal information that directly constrains or
↪  clarifies the answer.
– Ignore vague, irrelevant, or background time references.
– Each extracted item must represent a clear, factual temporal statement.

**Input Format:**
The `<memories>` section contains a list of `<memory>` tags.
Each memory is formatted as:
```
<memory [date_time="..." session_id="..." dia_id="..."]>
memory content text
</memory>
```
Where:
– `date_time`, `session_id`, `dia_id` are optional metadata attributes.
– The text within the tags is the content to analyze for temporal
↪  information.

To complete the task systematically, please follow the steps reasoning
↪  framework outlined below:

**Reasoning Steps:**
**PLAN:**
– Analyze the query's intent and define what constitutes relevant temporal
↪  information.
– Identify key temporal dimensions (e.g., dates, durations, sequences,
↪  constraints).

**ACT:**
– Extract temporal facts from each memory based on the defined criteria.
– For each fact, briefly justify its relevance to the query.

**REFLECT:**
– Review all extracted facts for consistency and query relevance.
– Make adjustments if needed to ensure factual support and coherence.

**Output Format:**
Your entire response must be a JSON array of temporal fact strings,
↪  formatted as follows:
`<answer>["temporal_fact1", "temporal_fact2", ...]</answer>`
Each string should clearly express a temporal fact (e.g., "Event occurred in
↪  March 2023", "Process lasted 3 days", "A happened before B").
If no relevant temporal information is found, output: `<answer>[]</answer>`

**Input:**
<query>{query}</query>
<memories>{memories_text}</memories>

## C.8. TOPIC MODULE RELATION PROMPT

---

TOPIC MODULE RELATION PROMPT (Low)

---

**Role:** You are a specialist in topic relationship extraction.
**Task:** From the provided memories, extract **only** topic relationships
↪   that are central to answering the given query.

**Extraction Criteria:**
– Focus on thematic connections and topic transitions that help understand
↪   the conversation flow.
– Identify main topics discussed and how they relate to each other.
– Extract topic shifts, topic continuations, and thematic relationships.

**Input Format:**
The `<memories>` section contains a list of `<memory>` tags.
Each memory is formatted as:
```
<memory [date_time="..." session_id="..." dia_id="..."]>
memory content text
</memory>
```
Where:
– `date_time`, `session_id`, `dia_id` are optional metadata attributes.
– The text within the tags is the content to analyze for topic
↪   relationships.

**Output Format:**
Your entire response must be a JSON array of topic relationship strings,
↪   formatted as follows:
`<answer>["topic_relationship1", "topic_relationship2", ...]</answer>`
Each string should clearly express a topic relationship (e.g., "Topic A
↪   leads to Topic B", "Discussion shifts from X to Y", "Topic C is related
↪   to Topic D through Z").
If no relevant topic relationships are found, output: `<answer>[]</answer>`

**Input:**
<query>{query}</query>
<memories>{memories_text}</memories>

---

## TOPIC MODULE RELATION PROMPT (Mid)

**Role:** You are a specialist in topic relationship extraction.
**Task:** From the provided memories, extract **only** topic relationships
↪  that are central to answering the given query.

**Extraction Criteria:**
– Focus on thematic connections and topic transitions that help understand
↪  the conversation flow.
– Identify main topics discussed and how they relate to each other.
– Extract topic shifts, topic continuations, and thematic relationships.

**Input Format:**
The `<memories>` section contains a list of `<memory>` tags.
Each memory is formatted as:
```
<memory [date_time="..." session_id="..." dia_id="..."]>
memory content text
</memory>
```
Where:
– `date_time`, `session_id`, `dia_id` are optional metadata attributes.
– The text within the tags is the content to analyze for topic
↪  relationships.

**Output Format:**
Your entire response must be a JSON array of topic relationship strings,
↪  formatted as follows:
`<answer>["topic_relationship1", "topic_relationship2", ...]</answer>`
Each string should clearly express a topic relationship (e.g., "Topic A
↪  leads to Topic B", "Discussion shifts from X to Y", "Topic C is related
↪  to Topic D through Z").
If no relevant topic relationships are found, output: `<answer>[]</answer>`

Let's think step by step to extract the topic relationships.

**Input:**
<query>{query}</query>
<memories>{memories_text}</memories>

## TOPIC MODULE RELATION PROMPT (High)

**Role:** You are a specialist in topic relationship extraction.
**Task:** From the provided memories, extract **only** topic relationships
↪  that are central to answering the given query.

**Extraction Criteria:**
- Focus on thematic connections and topic transitions that help understand
↪  the conversation flow.
- Identify main topics discussed and how they relate to each other.
- Extract topic shifts, topic continuations, and thematic relationships.

**Input Format:**
The `<memories>` section contains a list of `<memory>` tags.
Each memory is formatted as:
```
<memory [date_time="..." session_id="..." dia_id="..."]>
memory content text
</memory>
```
Where:
- `date_time`, `session_id`, `dia_id` are optional metadata attributes.
- The text within the tags is the content to analyze for topic
↪  relationships.

To complete the task systematically, please follow the steps reasoning
↪  framework outlined below:

**Reasoning Steps:**
**PLAN:**
- Analyze the query's intent to determine what topic relationships are
↪  relevant.
- Identify key thematic elements and potential connections.

**ACT:**
- Extract topic relationships from each memory based on content analysis.
- Identify thematic shifts, continuations, and interconnections.

**REFLECT:**
- Review all extracted relationships for relevance to the query.
- Ensure relationships are clearly supported by the memory content.

**Output Format:**
First write your reasoning following the PLAN → ACT → REFLECT structure.
Then, your final response must be **only** the following line:
`<answer>["topic_relationship1", "topic_relationship2", ...]</answer>`
Each string should clearly express a topic relationship (e.g., "Topic A
↪  leads to Topic B", "Discussion shifts from X to Y", "Topic C is related
↪  to Topic D through Z").
If no relevant topic relationships are found, output: `<answer>[]</answer>`

**Input:**
<query>{query}</query>
<memories>{memories_text}</memories>

## C.9. SUMMARY MODULE PROMPT

SUMMARY MODULE PROMPT (Low)

```
**Role:** You are a specialist in information synthesis.
**Task:** Based strictly on the provided entity, temporal, and topic
↪   relations, integrate, extract, and reorganize this knowledge to create a
↪   concise summary that clearly highlights the most useful information for
↪   answering the query.

**Input Format:**
The input includes:
1. A `<query>` defining the subject and scope.
2. An `<Entity Relations>` section containing one `<entity>` tag per
↪   relationship string.
3. A `<Temporal Relations>` section containing one `<temporal>` tag per
↪   temporal fact.
4. A `<Topic Relations>` section containing one `<topic>` tag per topic
↪   relationship.

**Synthesis Guidelines:**
- Do **not** answer the query directly.
- Explain what information is available and how it should be used to
↪   formulate an answer.

**Output Format:**
Your entire response must end with the following line:
`<answer>your summary text here</answer>`
The content inside `<answer>` must be plain text.

**Input:**
<query>{query}</query>
<Entity Relations>{entity_text}</Entity Relations>
<Temporal Relations>{temporal_text}</Temporal Relations>
<Topic Relations>{topic_text}</Topic Relations>
```

## SUMMARY MODULE PROMPT (Mid)

**Role:** You are a specialist in information synthesis.
**Task:** Based strictly on the provided entity, temporal, and topic
↪   relations, integrate, extract, and reorganize this knowledge to create a
↪   concise summary that clearly highlights the most useful information for
↪   answering the query.

**Input Format:**
The input includes:
1. A `<query>` defining the subject and scope.
2. An `<Entity Relations>` section containing one `<entity>` tag per
↪   relationship string.
3. A `<Temporal Relations>` section containing one `<temporal>` tag per
↪   temporal fact.
4. A `<Topic Relations>` section containing one `<topic>` tag per topic
↪   relationship.

**Synthesis Guidelines:**
− **Integrate** relevant entity, temporal, and topic facts into a coherent
↪   structure.
− **Extract** key information that directly supports or constrains the
↪   answer.
− **Reorganize** content for clarity and logical flow.
− Do **not** answer the query directly.
− Explain what information is available and how it should be used to
↪   formulate an answer.

**Output Format:**
Your entire response must end with the following line:
`<answer>your summary text here</answer>`
The content inside `<answer>` must be plain text.

Let's think step by step to synthesize the summary.

**Input:**
<query>{query}</query>
<Entity Relations>{entity_text}</Entity Relations>
<Temporal Relations>{temporal_text}</Temporal Relations>
<Topic Relations>{topic_text}</Topic Relations>

SUMMARY MODULE PROMPT (High)

```
**Role:** You are a specialist in information synthesis.
**Task:** Based strictly on the provided entity, temporal, and topic
↪    relations, integrate, extract, and reorganize this knowledge to create a
↪    concise summary that clearly highlights the most useful information for
↪    answering the query.

**Input Format:**
The input includes:
1. A `<query>` defining the subject and scope.
2. An `<Entity Relations>` section containing one `<entity>` tag per
↪    relationship string.
3. An `<Temporal Relations>` section containing one `<temporal>` tag per
↪    temporal fact.
4. A `<Topic Relations>` section containing one `<topic>` tag per topic
↪    relationship.

**Synthesis Guidelines:**
- Do **not** answer the query directly.
- Explain what information is available and how it should be used to
↪    formulate an answer.

To complete the task systematically, please follow the steps reasoning
↪    framework outlined below:

**Reasoning Steps:**
**PLAN:**
- Analyze the query's intent to determine the core information requirements.
- Identify how to organize and prioritize the provided relations.

**ACT:**
- Integrate entity, temporal, and topic facts into a coherent structure.
- Extract and highlight key information most relevant to answering the
↪    query.

**REFLECT:**
- Review the summary for clarity, completeness, and focus on the query.
- Ensure the summary effectively explains how the information should be
↪    used.

**Output Format:**
First write your reasoning following the PLAN → ACT → REFLECT structure.
Then, your final response must be **only** the following line:
`<answer>your summary text here</answer>`
The content inside `<answer>` must be plain text.

**Input:**
<query>{query}</query>
<Entity Relations>{entity_text}</Entity Relations>
<Temporal Relations>{temporal_text}</Temporal Relations>
<Topic Relations>{topic_text}</Topic Relations>
```

