# OpenReview forum: "Learning Query-Aware Budget-Tier Routing for Runtime Agent Memory"
_ICML.cc/2026/Conference — ICML 2026 regular_

### Official Review · Reviewer_CKWi · 2026-03-10

**Soundness:** 3
**Presentation:** 3
**Significance:** 3
**Originality:** 3
**Overall Recommendation:** 4
**Confidence:** 2

**Summary:**

The paper introduces BudgetMem, a framework designed to manage runtime memory extraction for LLM agents by providing explicit performance-cost trade-offs. The authors argue that existing "build once, use always" offline memory pipelines are inefficient and inflexible. To address this, BudgetMem proposes a modular runtime pipeline where each module offers three budget tiers: LOW, MID, and HIGH. A shared, lightweight router—trained via Reinforcement Learning  with a cost-aware reward—dynamically selects the appropriate tier for each module as the query is processed. The study explores three orthogonal tiering strategies: implementation complexity, reasoning behavior, and model capacity. Experimental evaluations on LoCoMo, LongMemEval, and HotpotQA show that BudgetMem outperforms strong baselines in high-budget settings and traces smooth, controllable performance-cost frontiers under tighter budgets.

**Compliance With Llm Reviewing Policy:**

Affirmed.

**Final Justification:**

The author’s response has fully resolved my confusion, and I will maintain my score.

**Key Questions For Authors:**

see Weeknesses

**Strengths And Weaknesses:**

Strengths:
1.Addressing the performance-cost trade-off explicitly at runtime for LLM agent memory is a highly practical and relatively underexplored area. The critique of query-agnostic preprocessing causing irreversible information loss or wasted computation is well-founded.
Strong and Comprehensive Empirical Results: The framework consistently demonstrates F1 and LLM-Judge improvements over a diverse set of strong baselines across multiple datasets . The trade-off curves clearly validate that the router successfully controls costs without catastrophic performance drops.

2.The introduction of the variance-based reward-scale alignment factor  to balance the task reward  and cost reward is a simple but highly effective technical contribution. The ablation study effectively proves its necessity in preventing the policy from collapsing into a degenerate low-cost state .

Weakness:

1.The paper measures cost strictly by aggregating API token usage multiplied by monetary prices. However, in runtime systems, latency (time-to-first-token, total execution time) is often an equally, if not more, critical constraint. The sequential nature of routing through multiple modules could introduce latency bottlenecks that are not reflected in the pure monetary cost evaluation.

2. While the router dynamically selects tiers (LOW/MID/HIGH) within each module, the underlying pipeline sequence (Entity, Filter, Temporal, Summary, Topic) appears rigid . The paper does not clearly state whether the router can dynamically skip a module entirely if it is irrelevant to the query (e.g., bypassing the Temporal module for a purely factual query). If even the "LOW" tier incurs a non-zero processing footprint, forcing every query through a fixed multi-stage sequence limits the maximum possible efficiency of the system.

---

> ### Author Rebuttal · Authors · 2026-03-31
>
> Thanks for your valuable feedback. We appreciate your insights and would like to further address the concerns you raised about our work.
>
> ## Latency
>
> Our main cost metric targets monetary cost. To additionally assess latency, we report results under a controlled local deployment (since API latency is heavily affected by network and provider-side variance and thus unsuitable for fair comparison) with Qwen as a shared backbone, using the same hardware and inference stack across all methods. We report the absolute latency in this controlled setting, while primarily using it to compare relative pipeline overhead across methods. As shown below, BudgetMem exhibits the expected latency trade-off across different budget settings, and the retrieval/filter overhead is practically controllable. We also report latency under different batch sizes to show that the relative overhead remains stable across serving settings. BudgetMem mainly focuses on the systematic study of trade-offs in agent runtime memory, while system-level acceleration of the overall pipeline is orthogonal to our goal and left for future work.
>
>
> | Latency - LoCoMo (BATCH=1) | Avg. Offline Memory Construction Time (ms) | Avg. Total Inference Latency (ms) | Avg. Retrieval (i.e., Filter Module) Latency (ms) | Avg. GPU Time (ms) |
> |---|:---|:---|:---|:---|
> | MemoryOS | 40657 | 2842 | 1488 | - |
> | A-MEM | 26842 | 1449 | 49 | - |
> | LightMem | 9740 | 1662 | 62 | - |
> | | | | | |
> | Ours-IMP (λ = 0) | - | 3881 | 1176 | 46 |
> | Ours-IMP (λ = 0.3) | - | 2432 | 556 | 46 |
> | Ours-IMP (λ = 0.9) | - | 1167 | 46 | 46 |
> | | | | | |
> | Ours-REA (λ = 0) | - | 3678 | 1274 | 47 |
> | Ours-REA (λ = 0.3) | - | 3429 | 1216 | 45 |
> | Ours-REA (λ = 0.9) | - | 3318 | 1209 | 48 |
> | | | | | |
> | Ours-CAP (λ = 0) | - | 3461 | 1228 | 44 |
> | Ours-CAP (λ = 0.3) | - | 3058 | 1079 | 46 |
> | Ours-CAP (λ = 0.9) | - | 2853 | 976 | 52 |
>
> | Avg. Total Inference Latency (ms) | MemoryOS | A-MEM | LightMem | Ours-REA (λ = 0) |
> |---|---:|---:|---:|---:|
> | BS=1 | 2842 | 1449 | 1662 | 3678 |
> | BS=2 | 1633 | 845 | 1059 | 2295 |
> | BS=4 | 974 | 518 | 775 | 1327 |
> | BS=8 | 604 | 296 | 438 | 858 |
> | BS=16 | 387 | 154 | 269 | 547 |
> | BS=32 | 203 | 96 | 163 | 326 |
>
> ## Pipeline
>
> We would like to clarify that the pipeline sequence shown in the paper is mainly a representative instantiation for **studying runtime performance-cost trade-offs in agent memory (our main focus)**, rather than a rigid design choice. BudgetMem itself is not tied to a specific module sequence: the pipeline can be flexibly configured, and our routing framework can be applied to different initialized module sets/pipeline structures (**Sec. 4.1, Line 146-150**). We also provide additional results with modified pipeline configurations, where a module is added (capture causal relationships), further demonstrating the generality of our approach.
>
> | Module-change (LoCoMo) | Judge | Cost |
> |---|---:|---:|
> | A-MEM | 32.96 | 2.88 |
> | LightMem | 40.76 | 1.50 |
> | Ours-IMP (Default) | 50.32 | 1.80 |
> | Add New Module: Capture Causal Relationships | 50.64 | 1.93 |
>
> ## Module Skip
>
> This is a reasonable question. In the current paper, we intentionally focus on **tier routing and runtime trade-offs** under a fixed initialized pipeline. We also experimented with adding a **NOOP/skip** option, but found that it brought only marginal changes on the current benchmarks, so we removed it for simplicity. That said, we agree that dynamic module skipping is a meaningful direction, and we expect its benefit to be more evident in real-world settings than in relatively static academic benchmarks. We therefore leave this as future work and provide the additional NOOP results below for reference.
>
> | LOCOMO | λ=0 w/o SKIP | λ=0 w/ SKIP | λ=0.1 w/o SKIP | λ=0.1 w/ SKIP | λ=0.5 w/o SKIP | λ=0.5 w/ SKIP |
> |---|---|---|---|---|---|---|
> | Ours-IMP | Judge: 50.32, Cost: 1.80 | Judge: 50.24, Cost: 1.80 | Judge: 52.55, Cost: 1.26 | Judge: 52.74, Cost: 1.22 | Judge: 28.66, Cost: 0.48 | Judge: 28.24, Cost: 0.44 |
> | Ours-REA | Judge: 52.23, Cost: 2.90 | Judge: 52.19, Cost: 2.90 | Judge: 53.50, Cost: 2.72 | Judge: 52.93, Cost: 2.67 | Judge: 51.75, Cost: 2.34 | Judge: 51.97, Cost: 2.30 |
> | Ours-CAP | Judge: 54.62, Cost: 2.40 | Judge: 54.47, Cost: 2.40 | Judge: 57.64, Cost: 2.48 | Judge: 57.12, Cost: 2.44 | Judge: 40.76, Cost: 0.58 | Judge: 39.56, Cost: 0.54 |
>
> Thank you again for your constructive feedback, and we commit to enriching the content of the paper based on the above in the revised version. We look forward to further engaging in discussions to improve the quality and impact of our work.

---

> > ### Author Rebuttal · Reviewer_CKWi · 2026-04-05
> >
> > The author’s response has fully resolved my confusion, and I will maintain my score.

---

### Official Review · Reviewer_xZVM · 2026-03-12

**Soundness:** 2
**Presentation:** 3
**Significance:** 2
**Originality:** 3
**Overall Recommendation:** 4
**Confidence:** 3

**Summary:**

This paper proposes a runtime agent memory framework that addresses the performance–cost trade-off in memory extraction for LLM agents. The authors formulate the problem as a module-level budget-tier routing task and design unified LOW/MID/HIGH budget interfaces for each module in the memory pipeline. A reinforcement learning router is trained to dynamically select different budget tiers to balance performance and cost during inference. Experiments are conducted on LoCoMo, LongMemEval, and HotpotQA datasets, and the results show that the proposed approach can achieve favorable performance–cost trade-offs compared with several baselines.

**Compliance With Llm Reviewing Policy:**

Affirmed.

**Key Questions For Authors:**

1. Baseline Cost Accounting Protocol:
   Table 1 reports significant cost differences between the proposed method and several baselines. However, it is unclear whether the reported cost includes offline memory construction, retrieval stages, and answer generation steps. Since the main claim of the paper relies on performance–cost comparisons, a clearer description of the cost accounting protocol is necessary.

2. Router State Representation:
   In Section 4.3, the router state is described as a combination of query, module inputs, and module descriptors. However, the paper does not specify how these elements are encoded (e.g., embedding model, vector dimensions, concatenation strategy). More detailed information is necessary for reproducibility.

3. Module Implementation Details:
   The implementation details of the entity, temporal, and topic modules are largely deferred to the appendix. Without a basic description of how the LOW/MID/HIGH tiers differ in implementation, it is difficult to determine whether the observed improvements mainly come from the routing strategy or from the module design itself.

4. Evaluation Protocol of LLM-as-a-Judge:
   The paper uses LLM-based judging for evaluation, but does not describe the prompt templates, inference parameters, or whether multiple runs were performed to reduce randomness. Please provide more details about the evaluation protocol.

5. Lack of Statistical Significance Analysis:
   The experimental results do not report standard deviations, confidence intervals, or results across multiple random seeds. Including statistical significance analysis would improve the credibility of the reported improvements.

6. Minor Issues:

The terms **“budget tier”** and **“tiering strategy”** are used interchangeably in different sections of the paper, but their hierarchical relationship is not clearly defined. The terminology should be made consistent throughout the manuscript.

The captions of **Figures 2–5** are relatively long and include interpretative conclusions. It is recommended that the captions only describe the content of the figures, while explanatory analysis should be moved to the main text.

**Limitations:**

yes

**Strengths And Weaknesses:**

Strengths:
Models the runtime memory cost–performance trade-off as a module-level budget-tier routing problem, allowing dynamic allocation of computational budgets across different stages of the memory pipeline.
Provides a modular memory framework with unified LOW/MID/HIGH budget interfaces, which enables flexible integration of heuristic, encoder-based, and LLM-based implementations.
Conducts relatively comprehensive experiments, including multi-dataset comparisons, cost–performance curves, reward ablation analysis, routing behavior analysis, and retrieval scale sensitivity analysis.

Weaknesses:
The cost modeling may be incomplete because the paper treats the cost of encoder-based modules as negligible, although such modules may still introduce non-trivial computational overhead in practical deployments.
Several experimental details are not sufficiently transparent, especially regarding cost accounting protocols and the evaluation procedure.

---

> ### Author Rebuttal · Authors · 2026-03-31
>
> Thanks for your valuable feedback. We appreciate your insights and would like to further address the concerns you raised about our work.
>
> ## Cost Modeling
>
> For encoder-based (non-LLM) operations, whose overhead is not well captured by token-based monetary cost, we additionally report their practical overhead through retrieval/filter latency under a controlled local deployment with Qwen as a shared backbone and the same hardware/inference stack across methods. As shown below, this overhead is non-zero but remains well controlled in practice, although it becomes higher when the budget is intentionally allocated to stronger retrieval. BudgetMem mainly focuses on the systematic study of trade-offs in agent runtime memory, while end-to-end system-level acceleration is left for future work. Due to strict rebuttal space limits, please kindly refer to our response to Reviewer Denm for full latency analysis. Thanks for your understanding!
>
> | Method | Retrieval / Filter Latency (ms) |
> |---|---:|
> | A-MEM | 49 |
> | LightMem | 62 |
> | Ours-IMP (λ = 0.9) | 46 |
> | Ours-IMP (λ = 0.3) | 556 |
> | Ours-IMP (λ = 0) | 1176 |
>
> ## Cost Accounting Protocol
>
> Cost is defined as the total token price of all LLM calls, including (1) offline memory construction, (2) LLM-based retrieval/memory extraction, and (3) query answering. We do not convert purely non-LLM retriever operations into token cost. The large cost of baselines mainly comes from processing the full raw interaction history with LLMs during offline memory construction, whereas BudgetMem performs more adaptive runtime memory extraction and thus avoids much of this unnecessary token usage.
>
>
> ## Router State Representation
>
> Specifically, we encode the query, module input, and module descriptor separately using all-mpnet-base-v2, producing three 768-d vectors. We then concatenate the query and module-input vectors and linearly project them to 256-d, and separately project the module-descriptor vector to 256-d. Finally, we concatenate these two 256-d representations to form the final 512-d state vector. We will clarify this in the revision and commit to releasing our code to further ensure reproducibility.
>
>
> ## Module Implementation Details
>
> Due to space limits, we have to move most module details to the appendix. At a high level, **implementation tiering** uses increasingly stronger implementations. For example, in the **entity module**, LOW uses rule-based extraction to identify basic entity mentions and relations, MID uses a BERT-based extractor for more robust structured extraction, and HIGH uses direct LLM-based extraction for the richest entity information. **Reasoning tiering** varies the reasoning paradigm (e.g., Direct / CoT / Reflection), and **capacity tiering** varies the underlying LLM size (**Line 199-230**). We will add a clearer basic description in the revised version.
> Regarding whether the gains come from routing or module design, module design and routing are inherently coupled in BudgetMem: the module design defines the available performance-cost tiers, while the router decides when to use each tier (i.e., higher tiers are designed and expected to offer stronger capability at higher budget.). Thus, the key idea of BudgetMem is to **systematically study runtime trade-offs** through routing among these tiers.
>
>
> ## Evaluation Protocol
>
> The LLM-judge prompt is already provided in **Appendix C.4 (Page 22)**. We will further clarify the inference details in the revised version: we use max tokens = 1024, temperature = 0, and report the average over 3 runs to reduce randomness in LLM judge evaluation.
>
>
> ## Statistical Significance
>
> In fact, we run 3 trials with different random seeds and record the variance, but do not include the full statistics in the paper due to space limits. Given the rebuttal space constraint, we present part of these results here and will include a more complete statistical analysis in the revised version.
>
> | LoCoMo (LLaMA) | F1 | Judge | Cost |
> |---|---:|---:|---:|
> | Ours-IMP | 38.75 ± 0.18 | 50.32 ± 0.14 | 1.80 ± 0.02 |
> | Ours-REA | 40.92 ± 0.10 | 52.23 ± 0.09 | 2.90 ± 0.01 |
> | Ours-CAP | 43.05 ± 0.08 | 54.62 ± 0.15 | 2.40 ± 0.01 |
>
> ## Minor Issues
>
> The two terms refer to different levels: **budget tier** denotes the three runtime choices **LOW/MID/HIGH**, while **tiering strategy** denotes the three ways of instantiating tiers, namely **implementation / reasoning / capacity**. In BudgetMem, we first initialize the system with one tiering strategy, and each strategy then exposes three budget tiers for routing (**Sec. 4.2, Line 187**). We apologize for the confusion and will make this hierarchy clearer and the terminology fully consistent in the revised version. We will also shorten the captions of Figures 2–5 and move interpretative discussion into the main text.
>
> Thank you again for your constructive feedback, and we look forward to further engaging in discussions to improve the quality and impact of our work.

---

> > ### Author Rebuttal · Reviewer_xZVM · 2026-04-01
> >
> > Thanks for the author, I have no further questions.

---

### Official Review · Reviewer_igSy · 2026-03-13

**Soundness:** 3
**Presentation:** 3
**Significance:** 3
**Originality:** 3
**Overall Recommendation:** 4
**Confidence:** 3

**Summary:**

This paper introduces BudgetMem, a runtime agent memory extraction framework. It breaks extraction into modules with three tiers named LOW MID HIGH, and trains a small PPO router to pick tiers per query and per module to control the quality cost tradeoff. It evaluates on LoCoMo LongMemEval and HotpotQA using F1, an LLM judge score, and token price based cost.

**Compliance With Llm Reviewing Policy:**

Affirmed.

**Key Questions For Authors:**

Dollar cost depends on external pricing, so results can change as prices change.
API Cost is just money. The actual availability of the system depends on time. The concurrency and serial latency of multiple calls to the Model need to be reported.

**Limitations:**

see weakness

**Strengths And Weaknesses:**

Pros:
- The modular pipeline and tier interface are easy to follow.
- It is nice to compare implementation reasoning and capacity tiering in one framework.
- The experiments cover three benchmarks and report both quality and cost.

Cons:
- PPO routing feels heavy unless you also compare against simple routers that use the same state, like a classifier or a greedy rule.
-Cost is defined as API dollars, but latency is missing. A multi stage pipeline can use fewer tokens yet feel slower because it makes several sequential calls and increases time to first token.
-Reward scale alignment and moving window normalization feel ad hoc. They make the reward drift during training, so it is hard to tell if PPO learns a robust policy or just fits the shaping.

---

> ### Author Rebuttal · Authors · 2026-03-31
>
> Thanks for your valuable feedback. We appreciate your insights and would like to further address the concerns you raised about our work.
>
> ## PPO Routing
>
> We use PPO because BudgetMem optimizes a performance-cost trade-off over **sequential discrete per-module tier decisions**, where gradients cannot be directly back-propagated through the routing process. Importantly, this does **not** make the method heavy: compared with greedy rules (e.g., always-low or always-high tier selection), BudgetMem adds only a small MLP router, so the extra parameters and GPU memory are all negligible in practice (GPU memory is dominated by the fixed embedding model). In this sense, the router is essentially lightweight like a small classifier, but PPO enables it to make **adaptive budget-tier decisions** rather than fixed greedy choices. This added flexibility is what allows BudgetMem to achieve substantially better trade-offs (as shown in Figure 2).
>
> | LoCoMo | Router Parameters (M) | GPU Memory (M) |
> |---|---:|---:|
> | Greedy Rule | 0 | 1268 |
> | BudgetMem | 0.86 | 1288 |
>
> ## Latency
>
> Our main cost metric targets monetary cost. To additionally assess latency, we report results under a controlled local deployment (since API latency is heavily affected by network and provider-side variance and thus unsuitable for fair comparison) with Qwen as a shared backbone, using the same hardware and inference stack across all methods. We report the absolute latency in this controlled setting, while primarily using it to compare relative pipeline overhead across methods. As shown below, BudgetMem exhibits the expected latency trade-off across different budget settings, and the retrieval/filter overhead is practically controllable. We also report latency under different batch sizes to show that the relative overhead remains stable across serving settings. BudgetMem mainly focuses on the systematic study of trade-offs in agent runtime memory, while system-level acceleration of the overall pipeline is orthogonal to our goal and left for future work.
>
>
> | Latency - LoCoMo (BATCH=1) | Avg. Offline Memory Construction Time (ms) | Avg. Total Inference Latency (ms) | Avg. Retrieval (i.e., Filter Module) Latency (ms) | Avg. GPU Time (ms) |
> |---|:---|:---|:---|:---|
> | MemoryOS | 40657 | 2842 | 1488 | - |
> | A-MEM | 26842 | 1449 | 49 | - |
> | LightMem | 9740 | 1662 | 62 | - |
> | | | | | |
> | Ours-IMP (λ = 0) | - | 3881 | 1176 | 46 |
> | Ours-IMP (λ = 0.3) | - | 2432 | 556 | 46 |
> | Ours-IMP (λ = 0.9) | - | 1167 | 46 | 46 |
> | | | | | |
> | Ours-REA (λ = 0) | - | 3678 | 1274 | 47 |
> | Ours-REA (λ = 0.3) | - | 3429 | 1216 | 45 |
> | Ours-REA (λ = 0.9) | - | 3318 | 1209 | 48 |
> | | | | | |
> | Ours-CAP (λ = 0) | - | 3461 | 1228 | 44 |
> | Ours-CAP (λ = 0.3) | - | 3058 | 1079 | 46 |
> | Ours-CAP (λ = 0.9) | - | 2853 | 976 | 52 |
>
> | Avg. Total Inference Latency (ms) | MemoryOS | A-MEM | LightMem | Ours-REA (λ = 0) |
> |---|---:|---:|---:|---:|
> | BS=1 | 2842 | 1449 | 1662 | 3678 |
> | BS=2 | 1633 | 845 | 1059 | 2295 |
> | BS=4 | 974 | 518 | 775 | 1327 |
> | BS=8 | 604 | 296 | 438 | 858 |
> | BS=16 | 387 | 154 | 269 | 547 |
> | BS=32 | 203 | 96 | 163 | 326 |
>
> ## Reward Design
>
> Reward-scale alignment is not an ad hoc design; it addresses a well-known stability issue in multi-objective RL[1,2], where balancing performance and cost is critical to avoid collapse toward degenerate policies. Our method provides a simple and effective way to stabilize this trade-off, which we hope can serve as a useful starting point for future work. Moreover, we additionally include cross-dataset generalization results using LLaMA under the perf-first setting below, which show that the learned router transfers well beyond the original training setting, suggesting that BudgetMem learns a generally effective policy rather than merely overfitting to reward shaping.
>
> | LOCOMO=>LongMemEval | Judge | Cost |
> |---|---:|---:|
> | A-MEM | 33.17 | 80.02 |
> | LightMem | 48.51 | 5.28 |
> | Ours-IMP | 56.60 | 0.71 |
> | Ours-REA | 57.00 | 0.68 |
> | Ours-CAP | 60.50 | 0.79 |
>
> [1] Router-R1: Teaching LLMs Multi-Round Routing and Aggregation via Reinforcement Learning. NeurIPS 2025
>
> [2] GDPO: Group reward-Decoupled Normalization Policy Optimization for Multi-reward RL Optimization. arXiv 2026
>
> Thank you again for your constructive feedback, and we look forward to further engaging in discussions to improve the quality and impact of our work.

---

### Official Review · Reviewer_Denm · 2026-03-17

**Soundness:** 3
**Presentation:** 4
**Significance:** 4
**Originality:** 3
**Overall Recommendation:** 4
**Confidence:** 2

**Summary:**

This paper proposes BudgetMem, which is a runtime agent memory framework that enables explicit, query-aware performance-cost control by structuring memory extraction into modules with different budget tiers and routing among them with a lightweight reinforcement-learned policy. BudgetMem consists of three tiering strategies, including implementation, reasoning, and capacity. Evaluation shows that BudgetMem improves accuracy-cost trade-offs across LoCoMo, LongMemEval, and HotpotQA benchmarks.

**Compliance With Llm Reviewing Policy:**

Affirmed.

**Key Questions For Authors:**

How well does a router trained on one dataset (e.g., LoCoMo) transfer to another (LongMemEval/HotpotQA) without retraining, and does transfer hold when the module set/pipeline changes?
Are there failure modes where higher tiers hurt (e.g., more reasoning increasing hallucination or overfitting to noise), and can you provide an error analysis tied to tier decisions per module?
How robust is the reward design across datasets, and how often do degenerate low-cost policies occur without careful tuning?

**Limitations:**

yes.

**Strengths And Weaknesses:**

The paper is well motivated on the problem concerning runtime memory extraction in a setting where performance vs. cost trade-offs must be made online per query. The paper also formalizes this problem in a reasonable way as as modulewise tier choice with a routing policy required. Treating memory extraction as a fixed pipeline whose modules expose low/mid/high tiers is a meaningful abstraction that makes budget control fine-grained and comparable across designs. Studying implementation , reasoning , and capacity tiering in the same backbone is one of new contributions as many prior systems mix these knobs without disentangling them.

However, the notion of budgeted routing might be incremental relateive to adaptive compute studies. The core idea on choosing heavier or lighter processing conditioned on the query/state sounds similar to existing conditional computation, mixture-of-experts gating, cascades, and agent reasoning level controls. In addition, the cost model treats non-LLM tiers as “negligible” and measures LLM cost by token pricing may not reflect real deployment bottlenecks, such as latency, retrieval cost, GPU time, batching effects. Especially, token pricing is somewhat not necessarilty closely copuled with the real deployment bottlenecks and may change depending on vendor business models and strategies.

The evaluation performs extensive experiments on task gains and cost-performance tradeoffs analysis compared to a broad set of basedlines. However, the evaluation may be further strengthened by investigating cross-dataset tradeoffs and robusteness, as well as generalization of the proposed design.

---

> ### Author Rebuttal · Authors · 2026-03-31
>
> Thanks for your valuable feedback. We appreciate your insights and would like to further address the concerns you raised about our work.
>
> ## Novelty
>
> BudgetMem is, to our knowledge, the first work to explicitly study controllable performance-cost trade-offs in agent runtime memory, which has been largely underexplored. We formulate this as a new runtime memory problem, propose a modular budgeted routing framework, and show how budget tiers can be effectively allocated across memory modules. This makes BudgetMem a distinct contribution to agent memory systems, rather than a direct extension of prior MoE, cascade, or conditional-compute paradigms, which are fundamentally different from ours.
>
> ## Latency
>
> Our main cost metric targets monetary cost. To additionally assess latency, we report results under a controlled local deployment (since API latency is heavily affected by network and provider-side variance and thus unsuitable for fair comparison) with Qwen as a shared backbone, using the same hardware and inference stack across all methods. We report the absolute latency in this controlled setting, while primarily using it to compare relative pipeline overhead across methods. As shown below, BudgetMem exhibits the expected latency trade-off across different budget settings, and the retrieval/filter overhead is practically controllable. We also report latency under different batch sizes to show that the relative overhead remains stable across serving settings. BudgetMem mainly focuses on the systematic study of trade-offs in agent runtime memory, while system-level acceleration of the overall pipeline is orthogonal to our goal and left for future work.
>
> | Latency - LoCoMo (BATCH=1) | Avg. Offline Memory Construction Time (ms) | Avg. Total Inference Latency (ms) | Avg. Retrieval (i.e., Filter Module) Latency (ms) | Avg. GPU Time (ms) |
> |---|:---|:---|:---|:---|
> | MemoryOS | 40657 | 2842 | 1488 | - |
> | A-MEM | 26842 | 1449 | 49 | - |
> | LightMem | 9740 | 1662 | 62 | - |
> | | | | | |
> | Ours-IMP (λ = 0) | - | 3881 | 1176 | 46 |
> | Ours-IMP (λ = 0.3) | - | 2432 | 556 | 46 |
> | Ours-IMP (λ = 0.9) | - | 1167 | 46 | 46 |
> | | | | | |
> | Ours-REA (λ = 0) | - | 3678 | 1274 | 47 |
> | Ours-REA (λ = 0.3) | - | 3429 | 1216 | 45 |
> | Ours-REA (λ = 0.9) | - | 3318 | 1209 | 48 |
> | | | | | |
> | Ours-CAP (λ = 0) | - | 3461 | 1228 | 44 |
> | Ours-CAP (λ = 0.3) | - | 3058 | 1079 | 46 |
> | Ours-CAP (λ = 0.9) | - | 2853 | 976 | 52 |
>
> | Avg. Total Inference Latency (ms) | MemoryOS | A-MEM | LightMem | Ours-REA (λ = 0) |
> |---|---:|---:|---:|---:|
> | BS=1 | 2842 | 1449 | 1662 | 3678 |
> | BS=2 | 1633 | 845 | 1059 | 2295 |
> | BS=4 | 974 | 518 | 775 | 1327 |
> | BS=8 | 604 | 296 | 438 | 858 |
> | BS=16 | 387 | 154 | 269 | 547 |
> | BS=32 | 203 | 96 | 163 | 326 |
>
> ## Transfer Exp
>
> We additionally include cross-dataset and module-change transfer experiments using LLaMA under the perf-first setting, as summarized in the following tables. The results show that BudgetMem maintains strong generalization under both settings and continues to significantly outperform strong baselines.
>
> | LOCOMO=>LongMemEval | Judge | Cost |
> |---|---:|---:|
> | A-MEM | 33.17 | 80.02 |
> | LightMem | 48.51 | 5.28 |
> | Ours-IMP | 56.60 | 0.71 |
> | Ours-REA | 57.00 | 0.68 |
> | Ours-CAP | 60.50 | 0.79 |
>
> | Module-change (LoCoMo) | Judge | Cost |
> |---|---:|---:|
> | A-MEM | 32.96 | 2.88 |
> | LightMem | 40.76 | 1.50 |
> | Ours-IMP (Default) | 50.32 | 1.80 |
> | Add New Module: Capture Causal Relationships | 50.64 | 1.93 |
>
> ## Error Analysis
>
> We additionally conduct a brief case analysis on LoCoMo. We do not observe a systematic pattern where higher tiers hurt. For example, on the simple query *“What kind of car does Evan drive?”*, both high- and low-tier settings recover the key fact (*Prius*). In contrast, for the more compositional query *“How many Prius has Evan owned?”*, the higher-tier setting extracts clearer multi-hop evidence, such as the old Prius breaking down and the new Prius being purchased later, which better supports the correct answer of *two Prius*. This suggests that higher tiers mainly help when richer relational/temporal evidence is needed, while lower tiers are often sufficient for simpler queries. We will further investigate this point in future work due to space limits.
>
> ## Reward Design
>
> We present additional trade-off results on LongMemEval, where the same reward design still achieves strong perf-cost trade-offs. Moreover, the proposed reward-scale alignment dynamically balances the performance and cost terms during training, reducing the need for careful tuning and helping avoid degenerate low-cost policies in practice.
>
> | LongMemEval | Judge | Cost |
> |---|---:|---:|
> | IMP(λ=0) | 56.00 | 0.71 |
> | IMP(λ=0.1) | 39.50 | 0.58 |
> | IMP(λ=0.9) | 39.00 | 0.58 |
> | REA(λ=0) | 58.00 | 0.67 |
> | REA(λ=0.1) | 55.00 | 0.62 |
> | REA(λ=0.9) | 54.00 | 0.61 |
> | CAP(λ=0) | 60.50 | 0.80 |
> | CAP(λ=0.1) | 48.50 | 0.13 |
> | CAP(λ=0.9) | 36.50 | 0.10 |
>
>
> Thank you again for your constructive feedback!

---

### Decision · Program_Chairs · 2026-04-30

**Decision:**

Accept (regular)

**Comment:**

This paper presents BudgetMem, a runtime agent memory framework that structures memory processing into modules with three budget tiers and trains a lightweight RL-based router to perform query-aware tier selection, balancing task performance against cost. All four reviewers gave scores of 4 (Weak Accept). The paper addresses a practical and underexplored problem with a clean formulation and solid experiments. I recommend weak accept.